# Mice in social conflict show rule-observance behavior enhancing long-term benefit

Il-Hwan Choe[1], Junweon Byun[1,2], Ko Keun Kim[1], Sol Park[1,3], Isaac Kim[1], Jaeseung Jeong[3] & Hee-Sup Shin[1,2]

Disorderly resolution of conflict is costly, whereas orderly resolution by consent rules enables quick settlement. However, it is unclear whether non-human animals can make and observe rules to resolve conflict without aggression. Here we report a new behavioral paradigm for mice: a modified two-armed maze that uses wireless electrical brain stimulation as reward. First, the mice were individually operant-trained to initiate and then receive the reward at the signaled arm. Next, two mice were coupled and had to cooperate to initiate reward but then to compete over reward allocation. Mice develop and observe a rule of reward zone allocation that increases the total amount of reward and reward equity between the pair. In the mutual rule-observance behavior, positive reciprocity and tolerance to the other's violation are also observed. These findings suggest that rodents can learn to make and observe rules to resolve conflict, enhancing long-term benefit and payoff equity.

[1] Center for Cognition and Sociality, Institute for Basic Science (IBS), Daejeon 34141, Korea. [2] IBS School, University of Science and Technology, Daejeon 34141, Korea. [3] Department of Bio and Brain Engineering, Korea Advanced Institute of Science and Technology (KAIST), Daejeon 34141, Korea. Correspondence and requests for materials should be addressed to H.-S.S. (email: shin@ibs.re.kr)

Social conflict occurs when the available resources are insufficient as animals compete to maximize individual benefit[1]. Competition is a common and natural strategy that nature favors[2], yet competition is costly[3] and stressful[4]. Costly fighting is largely continued until one party submits, and counterblows of the other are always risky. Worst of all, competitors may both suffer severe injuries. One party may give up early to save cost after rapid evaluation of the other's potential in battle, which may depend on size, appearance, experience and so on[5], but there is nothing to be gained in this scenario besides a likely safer exit. In this sense, disorderly competition is often wasteful in a society[6].

In contrast, the orderly resolution of conflict by making and observing rules (or conventions), could save costs and ultimately increase mutual benefits. These social rules include examples as the first to arrive is the first who is served/has the first choice, and respect of ownership[7]. In game theory, such conflict resolution has been called a 'Bourgeois' strategy[8]. In ecological systems, a Bourgeois strategy is found in some species displaying territorial ownership, e.g., butterfly, damselfly and social spiders[9–12]. In these species, when an individual finds a prior resident in a new territory, it retreats from the place, regardless of the resident's resource holding potential in battle. However, if it finds no resident, it occupies the place and then repels intruders, using aggression if necessary. Repeated interaction between two individuals using the Bourgeois strategy corresponds to mutual rule-observance. Such a strategy incurs little cost and distributes the resources equally, if the role of resident/intruder is determined stochastically during long-term interactions[7, 8], and can resolve conflicts quickly[12].

Evolutionary studies have suggested that natural selection favors individuals that use such strategies, thereby limiting aggression and saving cost[3]. Resolving conflicts by using rules is indeed often better than costly competition in terms of saving cost[7], and the Bourgeois strategy is one that can be dominant in certain populations[13, 14]. Humans, for example, are a species that utilizes the Bourgeois strategy, making and observing rules that are learned in the course of socialization[15]. These learned strategies can then be transmitted generation by generation; yet when the capacity to learn such rules evolved remains unknown[16]. Thus, it is not known whether fellow mammals, such as rodents, have the cognitive capability to develop the Bourgeois strategy and, if they do, how such rule observance behavior is spontaneously learned during social conflict over limited rewards.

Impulsivity has been suggested as a factor that prohibits animals from learning higher-level cooperative resolution[17–19]. Non-human animals are often impulsive and choose immediate, smaller rewards rather than waiting for larger, future rewards; this choice often leads them into potentially unnecessary conflict[19]. Impulsive animals therefore find it difficult to learn mutual rule-observance behavior, because it requires patience for potentially uncertain long-term profits. On the other hand, it has been argued that this observed impulsivity in animals is largely due to heightened food-deprivation[20]. Food is essential for survival and has been used as a primary reinforcement in animal experiments and, therefore, food-deprived animals become impulsive and tend to choose immediate rewards. Moreover, computer simulations suggest that cooperative self-reinforcing solutions can be produced in social conflict as long as the individuals involved possess a sufficient ability to learn such rules[21]. Put another way, individuals who are cognitively able to will develop and adopt simple behavioral rules, such as habits, rituals, routines and norms, when these rules are beneficial[22]. Yet it remains unclear whether non-human animals, such as mice, can spontaneously learn to adopt this Bourgeois strategy to save time, energy or other conflict-induced costs[12].

Here, we establish a new assay to investigate the emergence of interactive social behavior in mice. This behavioral paradigm required mice to first be trained on an operant conditioning paradigm in a two-armed maze, with wireless deep-brain stimulation into the medial forebrain bundle as a reward. Then, two mice were paired and had to share the same space to initiate the reward. However, the reward was only received by the mouse who reached the end of the arm, the reward zone, first, and could be disrupted by the entry of the other mouse into this same zone. Our results show that these mice settle the potential social conflict induced by this design by developing and observing the rule of reward zone allocation. More specifically, each mouse in the pair prefers one of the two reward zones, and lets their partner experience the reward in the non-preferred side. This behavior results in a maximization of the total amount of reward, as well as ensuring that the two mice are rewarded approximately equally. Taken together, these results suggest that mice in social conflict are able to develop and follow rules that enhance their long-term benefit.

## Results

**Wireless brain stimulation effectively trains mice to seek reward.** The aim of this study was to investigate whether or not mice can learn to make and observe rules that allow them to resolve conflict over limited rewards in an orderly fashion. To do so, we developed an operant system that utilizes wireless electrical brain stimulation (WBS) as reward (Fig. 1a and Supplementary Fig. 1). Electrical brain stimulation has been previously used in animal operant conditioning[23–25]; it has an incentive salience[24], and rarely induces satiation, unlike food[23]. Importantly, it does not require animals to be deprived of food, which is thought to make animals impulsive[20].

The WBS headset was small (1.5 × 1.5 cm) and lightweight (1.2 g), and generated an electrical current when it sensed an infrared signal from the external controller. The WBS headset was connected to a bipolar electrode that was implanted into a part of the reward circuitry in the brain, the medial forebrain bundle[24, 25] (Supplementary Fig. 2). We chose mice (C57BL/6J) as the subject species, because they possess measurable levels of social traits and learning ability[26, 27], and because they are a representative mammalian species.

First, we compared the conditioning efficacy and provocation of aggression between a WBS- and food-reward condition. Mice in the food reward condition were food-deprived (see Methods), while those in the WBS-reward condition were not. We conditioned individual mice in a two-armed maze that was operated in a self-directed manner, and thus enabled spontaneous learning and performance (Fig. 1b). Briefly, the two-armed maze consisted of three zones: a central zone (start zone); a left zone (reward zone); and a right zone (reward zone). A mouse could initiate a round by entering the start zone, which activated a visual cue (blue light) to randomly denote the reward zone, which could be either the left zone or the right zone, The frequency of the designated zones was counterbalanced (left zone: right zone = 0.5:0.5). We utilized a food-pellet (20 mg) for the food-reward condition and five-second WBS for the WBS-reward condition (Supplementary Movies 1 and 2).

The WBS reward was found to be a very effective positive reinforcement for the operant training, as shown by the steeper learning curve (Fig. 1c) and the faster movement toward the reward zone in response to the cue (Fig. 1d; one-way ANOVA on ranks, Dunn's correction, *$P < 0.05$, $n = 15, 50, 11$, for food-, WBS-, sham-WBS group, respectively), compared to those of the sham-WBS control group. Next, we selected the mice that had successfully obtained reward. These mice performed at 75% above

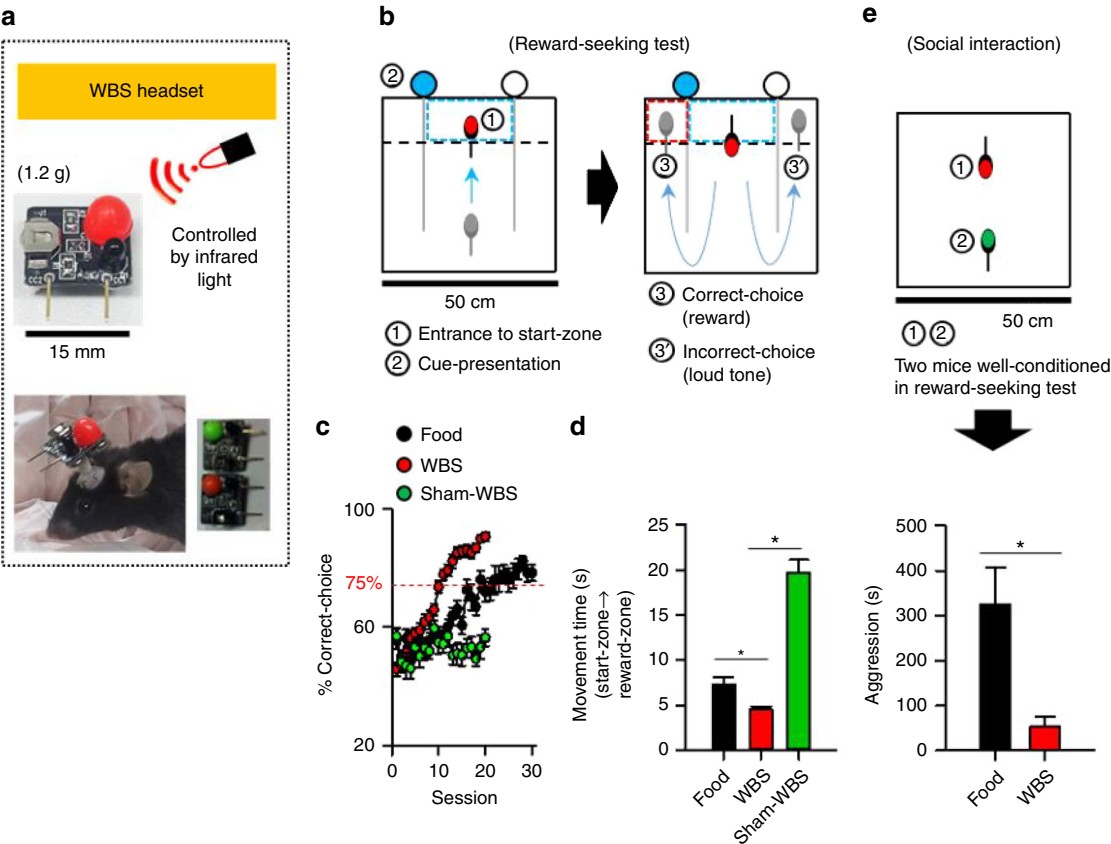

**Fig. 1** Wireless brain stimulation (WBS). **a** A small, lightweight WBS headset was used to stimulate a reward center, the medial forebrain bundle in the mouse. **b** Performance with WBS was tested in a two-way choice test. **c** Mice were conditioned rapidly when WBS was the reward. Red-dotted line indicates the significance level of the correct choice (binomial test, $P < 0.05$). **d** Mice moved quickly from start-zone to reward-zone when WBS was the reward (One-way analysis of variance on ranks, Dunn's correction, *$P < 0.05$). **e** The two mice in the WBS-reward condition displayed a shorter duration of aggression in a social interaction test than those in the food-reward condition (t-test, *$P < 0.05$)

chance (which was determined using a binomial test, with maximum trial number, 20; probability value, 0.5; criterion value, $P < 0.05$) and performance was based on the mean value of correct choice percentage throughout last three sessions. These criteria ensured that the mice were well trained. We then placed pairs (five pairs for food-reward group, 19 pairs for the WBS group) of well-conditioned mice in an open field for 30 min to observe their spontaneous aggressive interactions, such as chasing, biting, poking and mounting[28]. Pairs from the WBS condition showed shorter periods of aggression, whereas pairs from the food condition exhibited longer periods of aggression (Fig. 1e, t-test, *$P < 0.05$ and Supplementary Movies 3 and 4). These results indicate that, for this operant conditioning task, WBS is efficient as a reward and does not provoke much aggressive behavior

**Aggressive behavior is not shown in the mice over WBS reward.** To examine how two mice resolve conflict over limited resources, we conducted a 'conflict resolution test' (Fig. 2a). We put two well-conditioned mice in the same two-way maze, and motivated them to move quickly toward the denoted zone to obtain reward exclusively—i.e., based on winner-take-all paradigm. Briefly, pairs in the WBS condition could initiate a round only by entering the start-zone together. Then the visual-cue randomly denoted the reward-zone (counterbalanced between the two arms). WBS was immediately provided to any mouse that reached the denoted zone first. Unless the other mouse entered

this zone, the first-comer received WBS for 5 s (we call this an intact round). However, if the other, late arriving mouse also entered the zone, we instantly stopped WBS, thereby finishing the round (we call this a disrupted round).

Regardless of the outcome of a round (i.e., intact or disrupted), the two mice could start the next round by re-entering the central zone together. Each pair of mice could repeat rounds up to 40 times in a session (which lasted for 20 min), and a given pair of mice performed 20 sessions over 20 days, i.e., one session per day. The conflict resolution test in the food condition was structured similarly to that in the WBS condition, except that a round was finished right after the food pellet was dispensed. The food condition experiment had no separation of intact vs. disrupted round because it was physically impossible in this set-up.

In this conflict resolution test, we first observed that the number of rounds increased with experience in both conditions (WBS reward group, from $16.84 \pm 1.33$ rounds to $33.16 \pm 1.80$ rounds; Food reward group, from $14.00 \pm 1.14$ rounds to $33.80 \pm 3.06$ rounds, mean $\pm$ S.E.M., Fig. 2b), indicating that mice successfully learned how to initiate a round together. Next, we found that aggression was observed in 57% of sessions in the food condition, and only in 8% of the sessions in the WBS condition, and the amount of time showing aggression in the food condition was significantly longer than that of the WBS condition (Mann–Whitney Rank Sum Test, $P = 0.005$, Fig. 2c). In the food condition, for example, one dominant mouse occasionally pushed the submissive mouse to the start zone at the beginning of a run, or attacked the submissive mouse when the submissive mouse

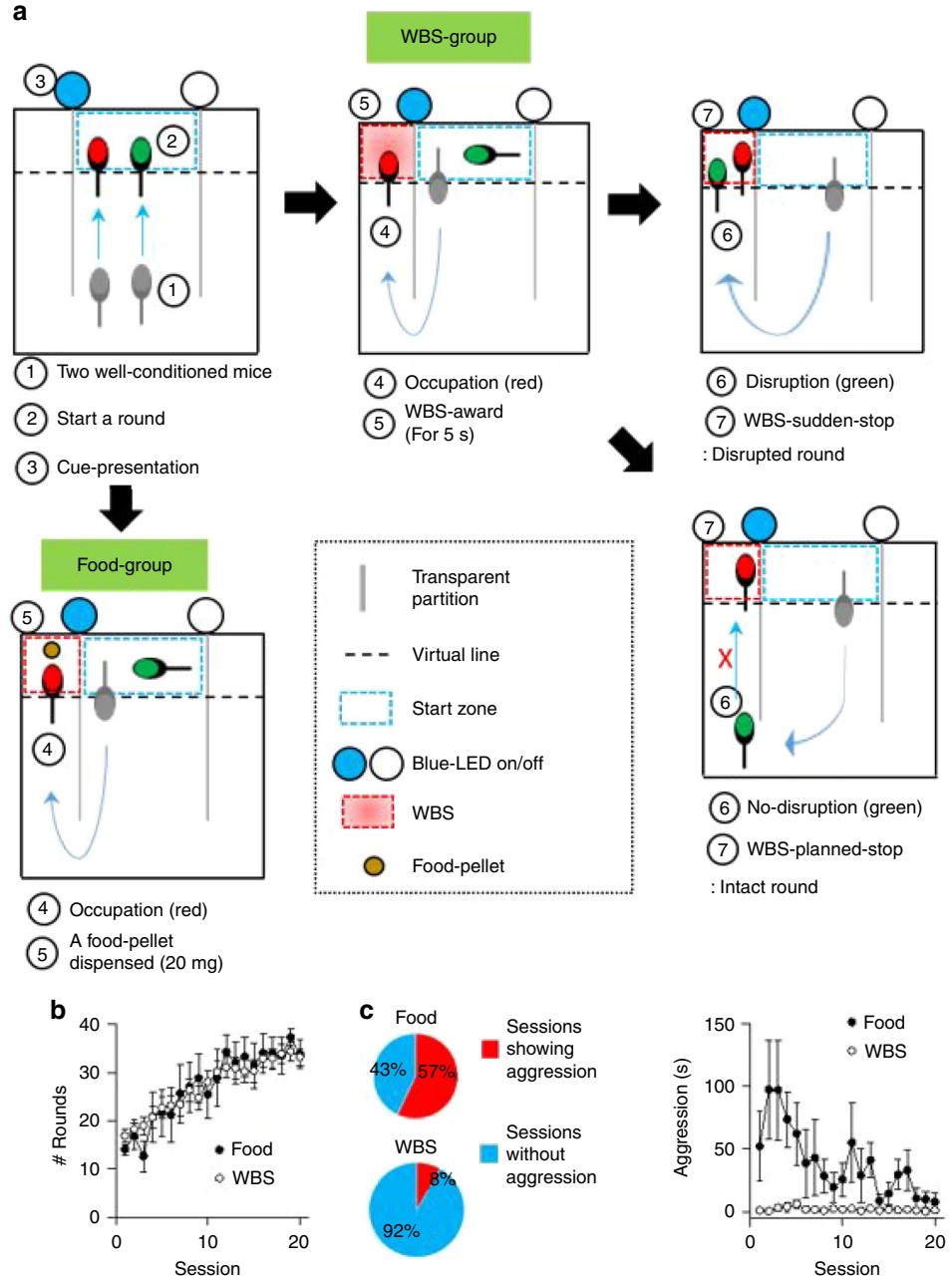

**Fig. 2** Less aggressive behavior in the WBS condition during the conflict resolution test. **a** Scheme of the conflict resolution test. A round was initiated when two well-conditioned mice entered the start-zone. Reward was provided when any mouse reached the zone denoted as the one with reward first. **b** The number of rounds increased over time in both conditions (Two-way RM analysis of variance (ANOVA), session, $F_{19, 418} = 19.3$, $P < 0.001$). **c** Mice in the WBS condition showed less aggression while mice in the food condition exhibited prominent aggression (two-way RM ANOVA, group X session, $F_{19, 418} = 8.1$, $P < 0.001$)

took the pellet (Supplementary Movie 5), resulting in the establishment of hierarchy and an unequal distribution of food (Supplementary Fig. 3). In the WBS condition, however, aggression was rarely observed (seen only in 31 sessions out of 380 sessions, and only occurring 47 times throughout the 31 sessions, Fig. 2c and Supplementary Movie 6). These observations suggest that the mice often resolved conflict over the limited rewards by aggression in the food condition, but did not use aggression to solve the conflict in the WBS condition.

**Mice develop and observe a rule of reward zone allocation**. How did the mice resolve conflict over the limited WBS without

aggression? To understand this better, first, we made pixel-based representation of the reward zone occupation rate by the two competing mice through the twenty sessions for each of the 19 pairs (Fig. 3a). In the early sessions, two mice in a pair competed for both of the reward zones. Eventually, however, the two mice showed a split behavior: when the left zone was denoted by the light cue, one mouse predominantly occupied the left zone. We called this mouse as $M_L$. When the light cue denoted the right zone, the other mouse (called $M_R$) predominantly occupied the right reward zone. As a result, 'reward zone allocation' was established. The time for establishing this reward zone allocation greatly varied among the pairs, with some pairs never reaching that level. For further analysis, we picked the pair #8, which

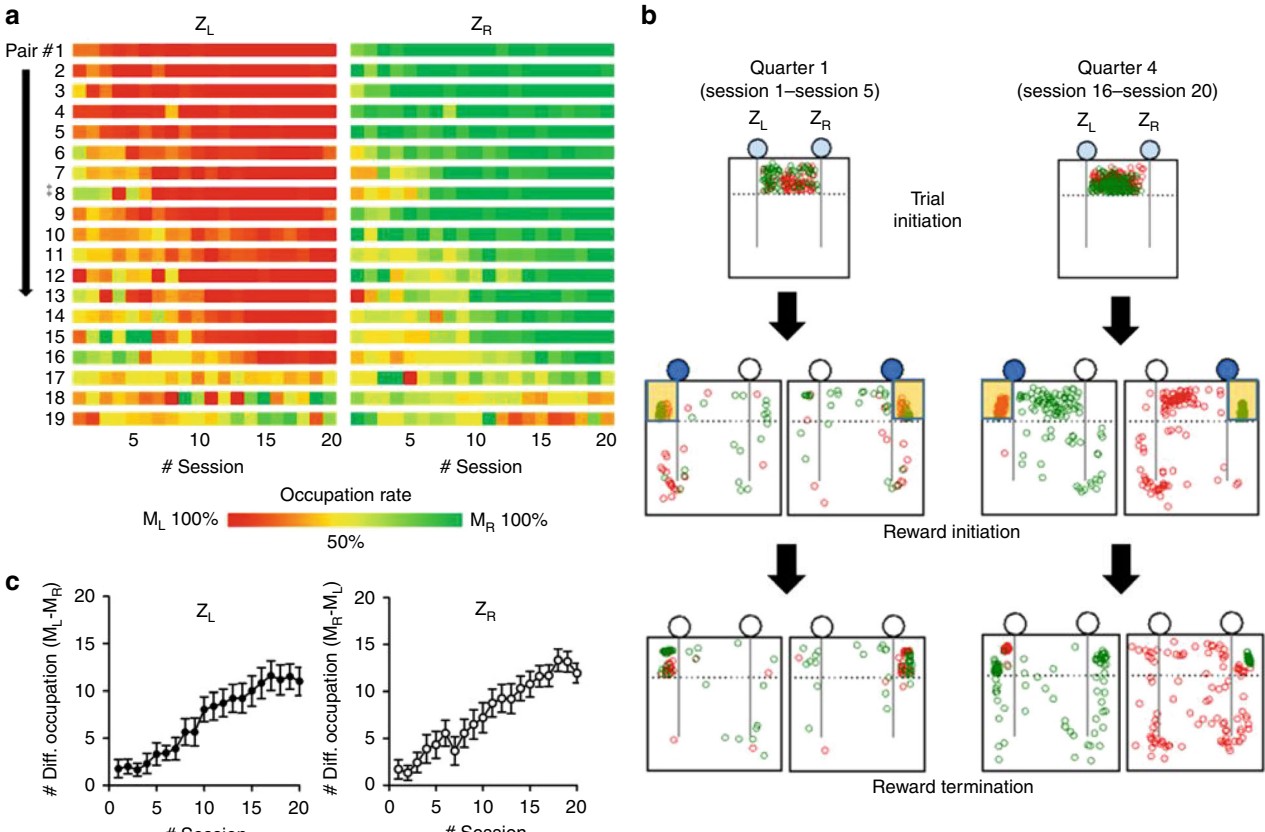

**Fig. 3** Development of behavioral patterns during conflict resolution test under the WBS conditions. **a** Pixel-based representation of the reward zone occupation rate by the two competing mice through the 20 sessions for each of the 19 pairs: Deep green, 100% by $M_R$; deep red, 100% by $M_L$; mixed colors representing different proportions by the two. $Z_L$, left reward zone, $Z_R$, right reward zone. **b** Pictorial presentation of the position of the mouse at given time points. Each circle (green, $M_R$; red, $M_L$) indicates the position of a mouse at the designated time point, trial initiation, reward initiation and reward termination. Left panel shows summation of the first quarter, from session 1 to 5. Right panel, the last quarter, from session 16–20. The ‡ eighth pair on the Fig. 3a is used as the representative sample here. **c** The mean value of the differential occupation of each reward zone by the two mice throughout the sessions. For $Z_L$, number of occupation by $M_L$—number of occupation by $M_R$. For $Z_R$, in the opposite direction

gradually developed the reward zone allocation, as a representative pair, and generated a space occupancy map for three different time points of a trial (Fig. 3b and Supplementary Fig. 7). We compared these space maps for the earlier sessions (Quarter 1, sessions 1–5) and later sessions (Quarter 4, sessions 16–20). This pictorial presentation of the occupancy map clearly showed that the reward zone allocation was well established in Quarter 4. Once their preferred zones were determined, when the left zone was lit up and taken by $M_L$, $M_R$ mostly stayed in the start-zone or in few cases ran toward its own side (the right side) which was unlit (Fig. 3b); $M_L$, the partner mouse, behaved similarly in rounds with the opposite situation. To show the gradual development of reward zone allocation, we plotted the mean value of the differential occupation of each reward zone by the two mice throughout the sessions. For $Z_L$, number of occupation by $M_L$—number of occupation by $M_R$. For $Z_R$, in the opposite direction (Fig. 3c). The two mice behaved as if they used the light cue for allocating the reward zones and taking turns in reward reception.

We defined this observed behavior of the mice as ‘rule-observance’: neither a preemptive-occupation nor a reward-disruption when the opponent received reward in its preferred zone. Preoccupancy or reward-disruption under the same condition was defined as ‘rule-violation’ (Fig. 4a). We successfully identified mice performing rule-observance ($M_{Obs}$) above chance level using the binomial test (maximum trial number, 40; probability value, 0.5; criterion value, $P < 0.05$). The

other mice were regarded as rule-violation mice ($M_{Vio}$, binomial test, $P > 0.05$).

One important prerequisite to conclude that this behavior is rule-observance rather than simple learning is to show that mice are capable of perceiving that the amount of reward is reduced when their competitor/partner enters the reward zone. To confirm that mice do indeed associate the presence of a conspecific with a decrease in reward, we carried out a control experiment with a modified protocol that was the same as the initial experiment except that the WBS reward was not discontinued when the competitor/partner mouse entered the reward zone (Supplementary Fig. 4a). The greatest difference in the behavioral pattern generated by these two protocols is observed at the reward termination, especially in the later sessions: Essentially all 40 trials end with the two mice within the reward zone (Supplementary Fig. 4b). Furthermore, only the pairs that went through the modified protocol (eight pairs) reached the maximum trial number per session (40 trials within 20 min) while none of the pairs in the main protocol (19 pairs) did (two-way repeated measure (RM) analysis of variance (ANOVA), $F_{1,25} = 6.681$, $P = 0.016$, Holm–Sidak post hoc test, $P < 0.05$, Supplementary Fig. 4c). These differences in the animals’ behavior between the two protocols suggest that, in the original protocol in which the presence of a conspecific could disrupt the reward, both mice were able to perceive that disrupting the others’ reward would result in a diminished

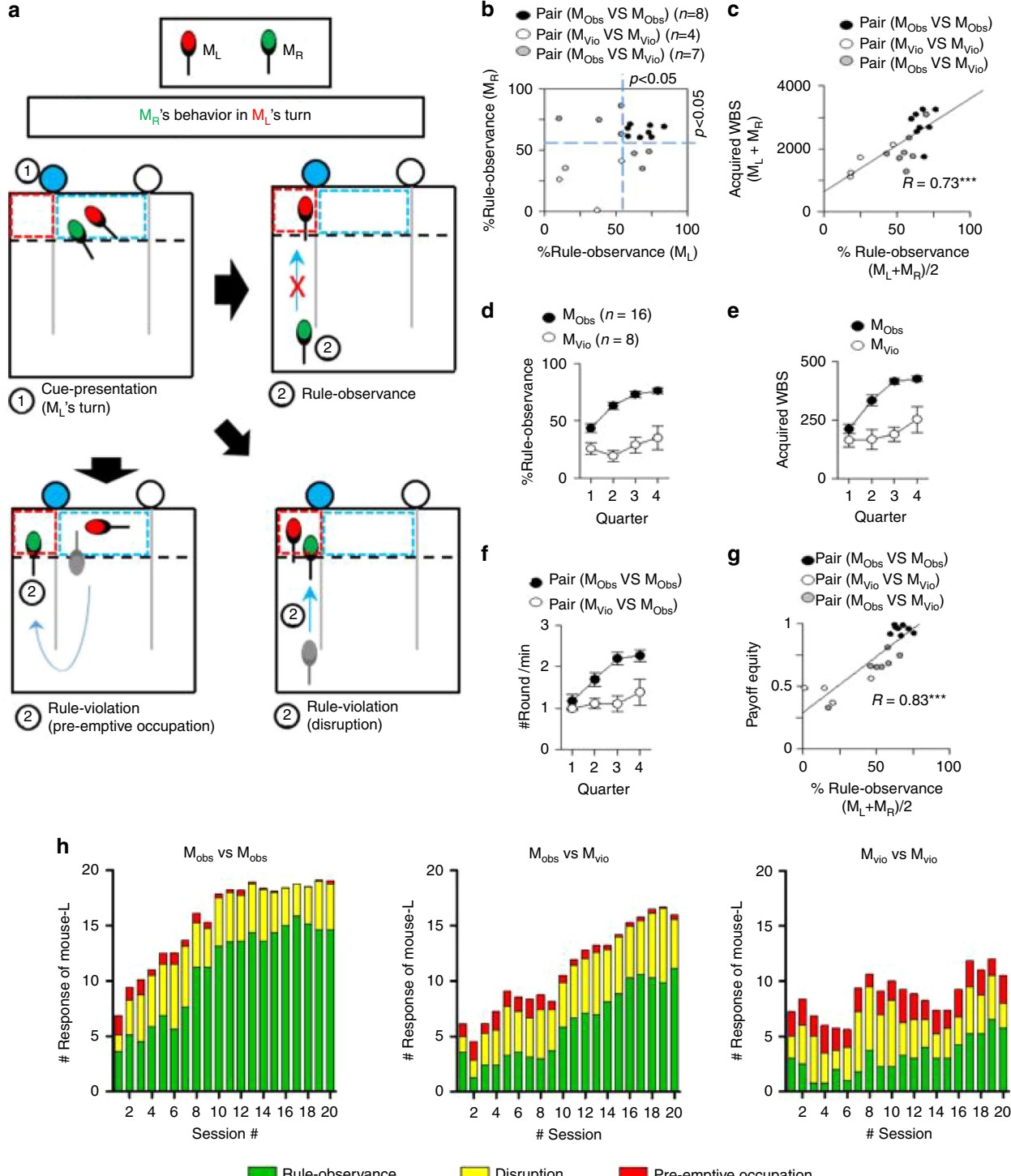

**Fig. 4** Mutual rule-observance strategy enhanced long-term benefit and payoff equity in the WBS condition. **a** Definition of three response types of $M_R$ when the left zone was light-cued ($M_L$'s turn): rule-observance, rule-violation (pre-emptive occupation) or rule-violation (disruption) behaviors. **b** The level of rule-observance behavior under the WBS conditions. Each dot represents % of rule-observance behavior of a pair. Blue reference lines indicates the significance level of rule-observance behavior (maximum trial number, 40; probability value, 0.5; criterion value, $P < 0.05$). $M_{Obs}$ is the mouse showing rule-observance behavior significantly and $M_{Vio}$ is the mouse showing rule-violation behavior. **c** Acquired WBS in the pair was plotted as the function of mean rule-observance of each pair. **d** Change in rule-observance behavior over time (filled circle, for mice in the $M_{Obs}$–$M_{Obs}$ pairs; open circle, for mice in the $M_{Vio}$–$M_{Vio}$ pairs). $M_{Obs}$ in the mutual rule-observance pair ($M_{Obs}$ and $M_{Obs}$) increased rule-observance behavior (one-way RM analysis of variance (ANOVA), quarter, $F_{3, 45} = 26.4$, $P < 0.001$) while $M_{Vio}$ in the mutual rule-violation pair ($M_{Vio}$ and $M_{Vio}$) did not. **e** Amount of acquired WBS was plotted over sessions: increased in $M_{Obs}$ (One-way RM ANOVA, quarter, $F_{3, 45} = 69.9$, $P < 0.001$) but not in $M_{Vio}$. **f** The number of rounds played per minute over sessions: increased in mutual rule-observance pairs (One-way RM ANOVA, quarter, $F_{3, 21} = 57.1$, $P < 0.001$) but not in mutual rule-violation pairs. **g** Payoff equity plotted against mean rule-observance for each pair in the three types of pairs: positive correlation between the two is observed. (***$P < 0.001$, Pearson's R). **h** Evolution of the response type of the $M_L$ through the sessions for each of the three pair types. Green bar, rule-observance; yellow bar, rule-violation (disruption); red bar, rule-violation (pre-emptive occupation)

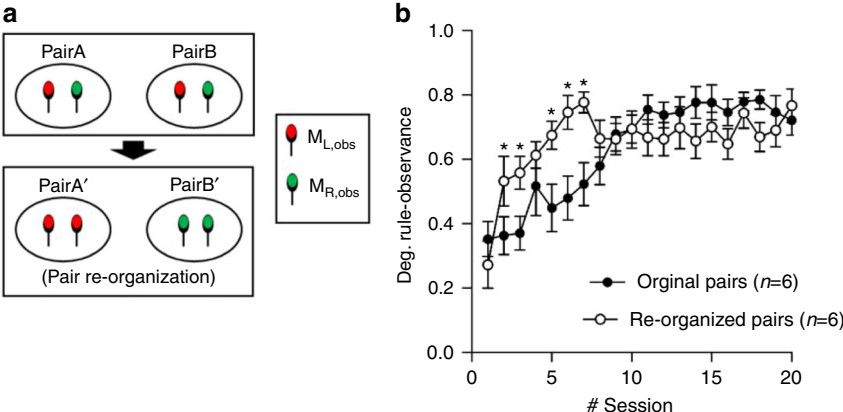

**Fig. 5** Rapid transfer of the reward allocation rule in the re-organized pairs. **a** From two $M_{Obs}$–$M_{Obs}$ pairs which have completed 20 sessions of the conflict resolution tests, the mice preferring the same side of the reward zone were chosen and subjected ($M_L$ with $M_L$, $M_R$ with $M_R$) to perform conflict resolution test for another 20 sessions. **b** Degree of rule-observance along the sessions. In the re-organized pairs, the degree of rule-observance more rapidly increased compared to the learning curve of the original pairs. The increase was the most dramatic between the first and the second session (two-way RM ANOVA, $F_{19,209} = 3.445$, $P < 0.001$, Holm–Sidak post hoc test *$P < 0.05$)

amount of the reward. This perception may have driven them to establish the rule observance behavior, which in turn allowed them to mutually enhance the amount of time each one got to experience the reward.

The proportion of $M_{Obs}$ was 60% of all mice (23/38). We plotted the level of rule-observance of $M_L$ and $M_R$ in a two-dimensional graph along with the binomial test result (Fig. 4b). This shows the presence of three separable sub-groups under the WBS conditions: mutual rule-observance pairs ($M_{Obs}$ and $M_{Obs}$), mutual rule-violation pairs ($M_{Vio}$ and $M_{Vio}$), and mixed pairs ($M_{Obs}$ and $M_{Vio}$).

**Rule-observance enhances long-term benefit and payoff equity.** Why did 60% of mice use the rule observance strategy to resolve conflict over limited rewards and what would be the potential advantages of the this rule observance vs. violating this rule? To address this issue, we investigated whether mutual rule-observance enhanced the amount of acquired reward (i.e., WBS) in a pair. We found that the degree of rule-observance in a pair was positively associated with the amount of acquired WBS in the pair (R = 0.73, ***$P < 0.001$, Pearsons' R, Fig. 4c). In addition, we compared $M_{Obs}$ in mutual rule-observance pairs with $M_{Vio}$ in mutual rule-violation pairs on the following parameters. $M_{Obs}$ clearly increased the frequency of rule-observance over time, indicating that rule-observance was learned in this group of mice, whereas $M_{Vio}$ did not (One-way RM ANOVA, quarter, $F_{3,45} = 26.4$, $P < 0.001$, Fig. 4d). The WBS acquisition in $M_{Obs}$, but not in $M_{Vio}$, showed a significant rise over training time, (One-way RM ANOVA, quarter, $F_{3,45} = 69.9$, $P < 0.001$, Fig. 4e). The rise of WBS acquisition in $M_{Obs}$ was likely due to an increase in the number of rounds they played. In fact, $M_{Obs}$–$M_{Obs}$ pairs took part in the increased number of rounds throughout the sessions and finally were able to participate in twice as many rounds $2.26 \pm 0.15$ rounds/min) as $M_{Vio}$–$M_{Vio}$ pairs ($1.39 \pm 0.32$ rounds/min) for a given period of time (One-way RM ANOVA, quarter, $F_{3,21} = 57.1$, $P < 0.001$, Fig. 4f).

In addition, we determined how the mutual rule observance strategy influenced the degree of payoff equity between the two mice participating this conflict resolution test. To do this, we calculated the reward acquisition ratio of the mouse that obtained less over the other mouse that obtained more. If two mice had acquired rewards equally during the test, the payoff equity value becomes close to 1. We found that the payoff equity value significantly increased from the third session ($0.52 \pm 0.07$, mean $\pm$ S.E.M.) to the 20th, final, session ($0.82 \pm 0.04$) (open circle, Friedman RM ANOVA on Ranks, $x^2 = 61.1$, d.f. = 19, $P < 0.001$, Supplementary Fig. 3). Moreover, the proportion of rule observance in a pair was positively associated with the payoff equity: the mutual rule-observance pairs achieved the highest level of payoff equity (>0.9) (R = 0.83, ***$P < 0.001$, Pearson's R, Fig. 4g). This finding strongly suggests that mutual rule-observance is an efficient way to achieve high payoff equity in conflict over limited rewards.

Rule violation behavior also evolved over the course of the sessions. A common pattern was observed in pair types that included $M_{obs}$. The mean value of the actual number of pre-emptive occupation decreased gradually ($1.16 \pm 0.19$ trials in the first quarter, $0.8 \pm 0.05$ trials in the last quarter) and the proportion of violation through disruption decreased (first quarter 35.19%, last quarter 19.27%), while rule-observance increased throughout the sessions ($5.2 \pm 0.50$ trials in the first quarter, $15.05 \pm 0.20$ trials in the last quarter, Fig. 4h and Supplementary Fig. 5).

Finally, we examined whether differences in traits between the two mice—including body weight, familiarity and learning ability —were associated with the level of payoff equity in the WBS condition; however, none of them showed a significant association with payoff equity (R = −0.26; Mann–Whitney Rank Sum Test, $P = 0.899$; R = −0.15, Supplementary Fig. 6).

**Position of the mouse pair during reward distribution.** Considering that rule-observance behavior requires one mouse to abstain from disrupting the other's reward (based on territory), the behavior of the mice who did not receive the reward while their partners receive reward is the key for establishing rule-observance. To understand the evolution of territory establishment in $M_{Obs}$–$M_{Obs}$ pairs compared to $M_{Vio}$–$M_{Vio}$ pairs, we analyzed the position of the mouse who did not receive the reward at the time point when the WBS reward was initiated or terminated to their partner (for the last five trials only; see Supplementary Table 2, Supplementary Fig. 7). At reward initiation in $M_{Obs}$–$M_{Obs}$ pairs, the majority of the opponents remained in the center zone ($86.0 \pm 2.5$%) and very few moved into the correct arm ($2.0 \pm 0.5$%). At reward termination, 5 s later, a large

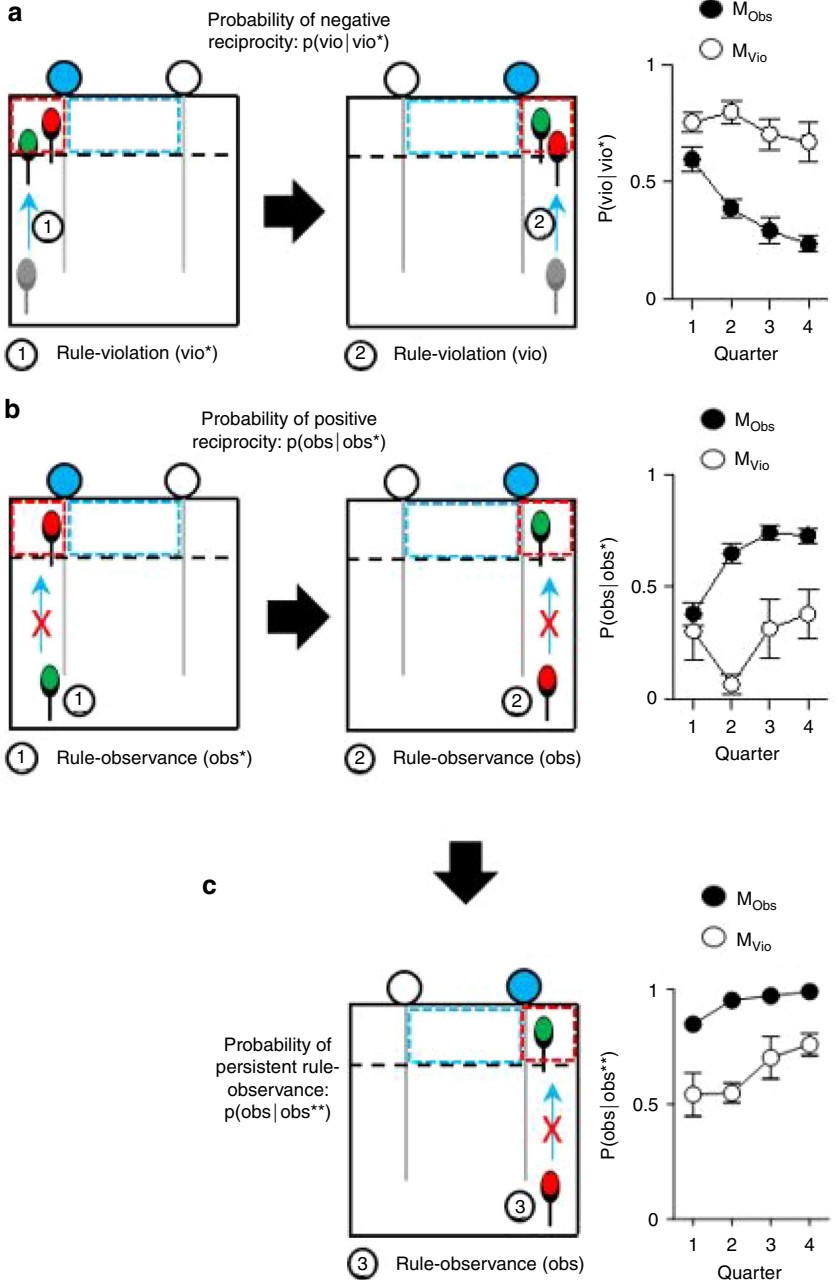

**Fig. 6** Reciprocity in mutual rule-observance. obs, the number of cases that the opponent mouse observed the rule during the partner's reward delivery in the current round. vio, the number of cases that the opponent mouse disrupted the partner's reward in the current round. obs*(or vio*), the number of cases that the opponent mouse showed rule-observance (or rule-violation) behavior in the previous round. obs**, The number of cases that the mouse performed positive reciprocity in the previous round. **a** Rule-violation of a mouse after the other's rule-violation in the previous round was defined as negative reciprocity. Probability of negative reciprocity decreased (i.e., tolerance increased) in $M_{Obs}$ (One-way RM ANOVA, quarter, $F_{3, 45} = 12.8$, $P < 0.001$) but not in $M_{Vio}$. **b** Rule-observance of a mouse after the other's rule-observance in the previous round was defined as positive reciprocity. Probability of positive reciprocity increased in $M_{Obs}$ (one-way RM ANOVA, quarter, $F_{3, 45} = 19.8$, $P < 0.001$) but not in $M_{Vio}$. **c** Rule-observance of a mouse after it already showed positive reciprocity in the previous round is defined as persistent rule-observance. Probability of persistent rule-observance at multiple unrewarded trials (i.e., partner rewarded.) was increased in both groups, but the level of $M_{Vio}$ remained lower than that of $M_{Obs}$ (two-way RM ANOVA, $F_{1,24} = 252.30$, $P < 0.0001$)

proportion of the mice who were staying in the center area moved out into the arms. Interestingly, the majority of them moved into the incorrect arm, thereby staying away from the correct arm in which their partner was receiving the reward. In contrast, the majority of opponents in the $M_{Vio}$–$M_{Vio}$ pairs at reward initiation time were in the correct arm ($56.4 \pm 15.1\%$). At the reward termination time, even more mice were positioned in the correct arm ($65.4 \pm 13.3\%$). Taken together, these results imply that the

rule-observant mice exerted an active effort not to disrupt their partner's reward.

**Mutual rule-observance is strategic not habitual**. To test whether mutual rule-observance was strategic or arose from a habitual preference for one side of the two-armed set up, we shuffled rodent pairs exhibiting mutual rule-observance: two $M_L$s (or

$M_{RS}$) were chosen from two different $M_{Obs}$–$M_{Obs}$ pairs and performed another 20 session of the conflict resolution test (Fig. 5a). In psychology and human neuroscience, a flexible and immediate adaption of one's behavior to a suddenly changed rule is called rapid rule-transfer[29]. In these re-organized pairs, the degree of rule-observance increased more rapidly compared to the learning curve of the original pairs (two-way RM ANOVA, $F_{19,209} = 3.445$, $P < 0.001$, Holm–Sidak post hoc test, $*P < 0.05$, Fig. 5b). As this is similar to rapid rule transfer in humans, we call this phenomenon rapid rule-transfer[29]. This finding suggests that the mice adopted mutual rule-observance due to strategic reasons rather than habits.

**Tolerance and reciprocity in mutual rule-observance behavior.** Despite the higher profitability of cooperative, rule-observance behavior, it is potentially vulnerable to violation or mistake. We investigated whether reciprocity was present in the mutual rule-observance strategy we observed in our mice. Mice may mirror each others' behaviors (i.e., tit-for-tat strategy)[30] or be tolerant of mistakes that their partner may make[31]. Reciprocity would best be shown in two successive rounds where the direction of cue alternated, (~60% of all rounds). In these sorted rounds, we estimated the negative reciprocity (p(vio|vio*)), that is, the number of cases in which one mouse showed rule-violation (vio) after the other mouse exhibited rule-violation in the previous round (vio*) over two successive rounds where the direction of cue alternated. This analysis revealed that $M_{Obs}$ showed a decreased p(vio|vio*) from $0.59 \pm 0.05$ in the beginning to $0.23 \pm 0.03$ to the end, i.e., an increased tolerance; $M_{Vio}$ on the other hand, exhibited a high level of p(vio|vio*) through training until the end ($M_{Obs}$ pairs, one-way RM ANOVA, quarter, $F_{3,45} = 12.8$, $P < 0.001$, Fig. 6a). This result shows that $M_{Obs}$ behaved tolerantly during their partner's reward even after a trial in which that same partner disrupted its own reward. This provides further evidence that these mice adopted a Bourgeois strategy.

We calculated the probability of positive reciprocity (p(obs| obs*)) using the same logic. We found that $M_{Obs}$ had increased p (obs|obs*) from $0.38 \pm 0.05$ in the beginning to $0.72 \pm 0.03$ at the end of training, whereas $M_{Vio}$ retained the low level of p(obs| obs*) throughout the trials ($M_{Obs}$ pairs, One-way RM ANOVA, quarter, $F_{3,45} = 19.8$, $P < 0.001$, Fig. 6b). Furthermore, to quantify the stability of the rule-observance behavior, we looked at the mice who showed positive reciprocity in a preceding unrewarded trial, to see how they behaved in an immediately following, unrewarded trial (obs**). The probability of persistent positive reciprocity (p(obs|obs**)) of $M_{Obs}$ was significantly higher than that of $M_{Vio}$ (two-way RM ANOVA, F1,24 = 252.30, $P < 0.0001$, Fig. 6c). This finding indicates that positive reciprocity behavior was stable in the mutual rule-observance pairs.

## Discussion

Non-human animals are thought to be impulsive, often choosing immediate reward even if it results in conflict, thereby failing to resolve potential social conflicts rationally[18, 19]. Here we have shown that mice find an orderly resolution to social conflict over limited rewards by making and observing the rule of 'reward zone allocation'. The current study further shows that this cooperative, rule-observance behavior is learned by both mice in a pair, thereby enhancing the long-term benefit and payoff equity for both mice. Thus, our study suggests that rule-observance behavior is a powerful, higher-level mechanism for conflict resolution, and suggests that mice can use it in addition to the well-established lower-level strategies, such as hierarchy, threat-display, ritual and war of attrition[5].

Research in game theory has shown that the orderly resolution of conflict by establishing and observing rules may eventually increase mutual benefits. Here, we see that the rule-observing mice gradually increased their tolerance and positive reciprocity towards their partners, while the mice who did not observe these rules also did not develop either tolerance or reciprocity. This rule-observing behavior seen in these mice may correspond to the Bourgeois strategy, as defined by classical game theory. Futhermore, we show that mice persistently observe this established rule even when the partner is rewarded in consecutive times, i.e., in a situation that is more costly for that mouse. Such results suggest that rule-observance behavior is stable once established. Considering the low positive reciprocity of $M_{Vio}$, it was surprising to see that the probability of persistent rule-observance behavior in $M_{Vio}$ was higher than 0.5 and even increased throughout the sessions. These $M_{Vio}$ mice represent a subgroup of the tested sample who showed rule-observance in the previous round. In other words, there is heterogeneity among $M_{Vio}$ in rule-observance/violation behavior. In addition, the total rule-observance behavior of $M_{Vio}$ increased slightly throughout the sessions. This suggests that $M_{Vio}$ may also be able to learn rule-observance, albeit at a much slower rate than $M_{Obs}$.

The novel paradigm we use to demonstrate this behavior should be developed further to allow for future studies on diverse cognitive/social behavioral questions, for example, to investigate how familiarity between a pair affects their rule-observance behavior, how the asymmetry of the amount of reward between the two arms affects the behavior, or what will happen to the rule-observance behavior of the trained mice when there is only one reward zone available. Moreover, diverse tools available for studies in the mouse should allow further research on the brain mechanisms underlying different stages of the behaviors involved in this assay.

In conclusion, here we show that mice in potential conflict over a limited reward can develop and observe the rule of reward zone allocation, thereby enhancing each individual mouse's benefit and payoff equity. These mice also show tolerance and positive reciprocity toward the partner's behavior, requiring active efforts not to disrupt their partner's reward.

## Methods

**Mice.** Male C57BL/6J mice were used for the current study. Four or five mice were housed together in a cage under a 12:12 light-dark cycle. During the time, food and water were accessible ad libitum. Mice were provided by the animal facilities in the Institute for Basic Science, Daejeon, Korea. All animal studies and experimental procedures were approved by the Animal Care and Use Committee of the Institute for Basic Science, Daejeon, Korea.

**Stereotaxic surgery.** At the 11th week in age, stereotaxic surgery to implant a bipolar electrode (MS303T/2-B/SPC, Plastics One, Roanoke, Virginia) onto the right medial forebrain bundle (+1.2; −1.2; −5, AP; ML; DV, in millimeter from the bregma) was performed. Ketamine (120 mg/kg) and xylazine (10 mg/kg) was administered to anesthetize mice before the stereotaxic surgery. After the electrode implantation, each mouse was housed alone until the end of behavioral test. Location of electrode was confirmed after sacrifice. After 1-week recovery from surgery, food was restricted in the food-group, which was randomly distributed among all mice. Care was given to keep mice body weight above 85% of the reference body weight, as measured 1 day before food-restriction. The reference body weight of mice was $27.6 \pm 0.41$ g (mean ± S.E.M.). In our animal care system, supplying food as much as 10% of reference body weight after a session of behavior test was sufficient to maintain target body weight. The average body weight during the food-restriction was $25.4 \pm 0.45$ g, equivalent to 92% of the reference body weight. For the WBS-group, ad libitum feeding was applied.

**Apparatus and materials.** The WBS system was primarily comprised of an infrared pulse emitter and a lightweight WBS-headset (1.2 g). First, a home-made electrical pulse generator sent pulse signal onto an infrared emitting diode (SIR-568ST3F, Rohm, Kyoto, Japan). Peak light emitting wavelength of the diode was 850 nm and the luminous output was 13 mW. Second, the WBS-headset sensed the infrared light signal through an infrared light sensitive phototransistor (RPT-

34PB3F, Rohm, Kyoto, Japan; 750–900 nm in spectral length). The WBS-headset transformed the sensed infrared light pulse into electrical pulse. Finally, the transformed electrical pulse was delivered to the medial forebrain bundle through the pre-implanted bipolar electrode. The headset was set to charge maximum 6.2 V onto brain tissue with impedance over 47 KΩ by installing a Zener diode (breakdown voltage, 6.2 V) and a resistor (R, 47 KΩ). Brain tissue impedance was over 47 KΩ in all mice. The average impedance was $115 \pm 4$ KΩ. In current study, we used five trains of infrared pulses to generate one time of WBS reward. Each train was generated every second. The number of pulses in a train was 30. A light pulse was given for 0.2 ms with 10-ms interval. The total length of a train was 0.3 s and resting time between two trains was 0.7 s. Corresponding to the light pulses, the WBS-headset generated five trains of electrical pulses. An individual electrical pulse was 1 ms long (from rise to half-decay) and 6.2 V in peak amplitude. The headset operated by a 12 V rechargeable battery pack (a serial connection of four ML-414, Panasonic, Japan). A red or green light-emitting diode (LED) -indicator was attachable onto the WBS-headset. We used LEDs to detect the location of mice.

**Behavioral study design**. In our operant conditioning box, a two-armed maze (50 cm × 50 cm × 30 cm, width × depth × height), a camera, an automatic color detection software, two blue LEDs, a speaker, two micro-pellet dispensers and the WBS-reward system were equipped. The two-armed maze was made by dividing an open field arena into three sections through the use of two transparent partitions. We virtually set a start zone in the central section (body) and two payoff zones in the left/right sections (two arms). The camera and the color detection software enabled us to monitor the location of the mouse inside the two-armed maze continuously. Through the location information, we controlled the function of components. Two blue LEDs were placed beside the payoff zones. Micro-pellet dispensers were also fixed adjacent to the payoff zones. The infrared pulse emitters for the WBS-reward system was installed on the ceiling of the operant conditioning box above 40 cm from the bottom.

We defined one trial as one chance to obtain a payoff. The trial was initiated when the freely moving mouse entered into the start zone. One of two blue LED was turned on to denote the correct-choice zone, where a positive reward would be supplied. If the mouse chose the denoted zone (correct choice), a micro-pellet (20 mg, F0163, BioServ, NJ, USA) was provided to mice in the food-group ($n = 15$). 5 s of WBS-reward was given to mice in the WBS-group ($n = 50$). During WBS-reward delivery, if the mouse leaves the reward zone and re-enters without disruption by the opponent, it would receive the stimulation for the remaining reward duration. For Sham-WBS-group, the headset was switched off, although the IR signal was provided as for WBS-reward group ($n = 11$). In the case that mice chose the unlit payoff zone (incorrect-choice zone), a negative reinforcement (loud tone, 75 dB, 0.5 s) was given. After the mouse received the reward, the trial was terminated and the operant conditioning system returned to idle state.

Unlike the freezing behavior demonstrated in response to fear, there are no simple behavioral markers for quantifying reward in mice. We can only tell whether mice prefer a longer reward to a shorter reward by comparing the time they visit the two reward zones with different duration of IR lighting time. In the preliminary experiments where we try to establish the stimulation reward protocol, we found that mice chose the longer over shorter stimulation reward: 6 s over 2 s ($n = 10$), or 6 s over 4 s ($n = 10$), conforming to 'the matching rule'. We could interpret these results that they can distinguish the different durations of the stimulus, and thus perceive them as different amounts of reward (Supplementary Table 1).

A daily training session allowed the mouse maximum 20 trials within 40 min. The denotation side was counterbalanced and randomized by pseudo-random sequences, which was designed to prevent four successive cues in a side. We prepared 10 sequences and each sequence was used every 10 days. This experimental system was fully automated from cue-presentation to rewarding to minimize the interruption by researchers during the test.

We evaluated the performance of each mouse by two criteria: a well-trained mouse should achieve the maximum number of trials during the last three sessions (60 trials) and the number of correct choices during that period should be over 45 (binomial test, 20 trials, 0.5 of probability, $P < 0.001$). In the food-group, mice were trained for 30 sessions and 11 out of 15 (73%) passed the criteria. In the WBS-group, 38 out of 50 (76%) passed the criteria in 20 sessions. In Sham-WBS-group, no mice passed this criterion.

Regarding pellet priming, the same micro-pellet was used before the operant conditioning. We put six micro-pellets in front of the two food-dispensers and exposed each mouse into the maze for 30 min. We repeated the priming until the mouse consumed all pellets inside the maze. Five days were sufficient for pellet priming.

In the WBS conditioning, almost all mice stayed in the correct-choice zone while the WBS was continuing. In the early period, some mice occasionally came out from the payoff zone while the WBS was continuing. That behavior usually disappeared in a few trials in well-trained mice. As a consequence of operant conditioning, five pairs for the food-group and 19 pairs for the WBS-group were prepared. In pairing, response time for reward, the time from the start zone to the payoff zone, was considered primarily. We averaged the response times in the last three sessions. By the value, all mice were ranked by descending order and we paired every two mice from above. In the food-group, the response time of 10 well-

trained mice was $7.3 \pm 0.84$ (mean $\pm$ S.E.M., in seconds). The difference in response time between mice in pair was $1.3 \pm 0.57$ s. In the WBS-group, the response time of 38 well-trained mice $4.4 \pm 0.40$ s. The difference in response time was $0.54 \pm 0.14$ s.

In the conflict resolution test, two well-trained mice were put in the same arena where those were trained. There are three additional rules in this test, as follows: First, the incorrect-choice condition was removed. If any mouse visited the unlit zone during a trial, no tone was given and the trial just continued. Second, a trial was initiated by a joint action, where two mice should be at the start zone together. Third, there was the disruption condition in WBS-group, in which one of the mice could disrupt their partners' reward by entering the reward zone. The mice were wearing one of two different colored headset (red or green) during the test. To measure the payoff, we counted the number of pellets each mouse consumed. Occasionally, the late mouse snatched the pellet that the first-comer was holding in its front paws. In such cases, the pellet was attributed to the late mouse. In the WBS-group, the time spent receiving the WBS in the payoff zone was measured. In the trials in which the other mouse stayed out the payoff zone, the rewarding time was 5 s. However, in the case of a disruption, only the duration from reward initiation to the onset of disruption was considered. Rarely, the first-comer temporarily left the payoff zone and re-entered within the maximum 5-second potential WBS-reward. In this case, we did not deduct the time that the first-comer was outside the reward zone.

To quantify the length of aggressive interactions, we measured the distance between two mice. We extracted video sections that the distance between two mice was less than 7 cm for over 3 s (first selections). Among the first selections, we manually identified sections that contained chasing, biting, pushing, poking or mounting (these we call second selections). In the case of two adjacent video sections that were thought to be fractions of a continuous action, we merged the two. Three observers participated in the manual second selection and were blinded to the experiment. To refine the second selections, three observers who did not participate in the behavioral experiment manually reconfirmed the video clips.

In WBS-group, seven pairs were made: two mice were housed together until 11th week in age, just before the surgery (familiar pairs). Twelve pairs were made with mice from different cages (unfamiliar pairs).

**Data analysis**. We analyzed our data using Matlab. There are no statistical methods for pre-determination of sample sizes, but we employed similar sample sizes to those that are generally accepted in the field. All statistical tests were non-parametric and two-tailed.

**Data availability**. The data supporting this study are available from the corresponding author for reasonable request. The computer code (MATLAB code) used for the analysis in this study are available from the corresponding author if the request is reasonable.

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

## Acknowledgements

This work was supported by the Institute for Basic Science (IBS-R001-D1).

## Author contributions

The system was set up by I.C. and K.K. I.C., K.K., S.P. and I.K. performed most of the experiments. Behavioral analysis was done by I.C., J.B., K.K., S.P. and I.K. The manuscript was prepared by I.C., J.B., J.J. and H.S.

## Additional information

**Competing interests:** The authors declare no competing financial interests.

