## [Peer Review File · Nature Communications]

Reviewers' comments:

Reviewer #1 (Remarks to the Author):

Review of Manuscript Number: NCOMMS-17-01018-T

Title: Mice in conflict show rule-observance behavior enhancing long-term benefit

The very interesting set of experiments described in this manuscript examined the behavior of mice learning to perform a two-armed maze task for either food reward or wireless brain stimulation (WBS) reward. The claim is that mice learned to observe social rules that allowed them to avoid conflict for mutual benefit and reward payoff equity. Although this claim may be true, the observed behavior also appears to be amenable to a simpler conditioning explanation. Thus, although the experiment is provocative, I cannot recommend publication until the authors can provide stronger evidence discounting the more parsimonious conditioning interpretation. Another concern is the small number of mouse pairs in the food reward condition, which made it difficult to interpret observed differences in learning between WBS and food reward conditions.

I. Conditioning interpretation of conflict resolution performance with WBS reward

An alternative to rule-observance is the idea that mice responded as they did due to discrimination learning. From this perspective, mice in the WBS reward condition learned to respond on cue to move to the left or right arm of the maze where they received WBS. WBS mice reinforced for responding to the left (ML) were never reinforced in the right arm of the maze, whereas mice reinforced for responding to the right (MR) were never reinforced in the left arm of the maze. Visual cues were associated with the appropriate goal when the correct mouse was present, so mice could learn that flashing lights in one location signaled WBS, whereas flashing lights in the opposite arm, signaling no WBS, were irrelevant. Critically, MR mice could not compete with ML mice for WBS in the left arm because no behavior – competitive or otherwise – could ever lead to a reinforcer in the incorrect arm. Furthermore, because there were no physical reinforcers involved in WBS, mice could not even observe other mice receiving a physical reinforcer. Thus, responses by WBS mice to the incorrect arm should not increase above some baseline level. The opposite would be experienced by MR and ML mice in the opposite (Right) arm of the maze. The predicted results would be rapid learning of distinct responses for left and right arms for both WBS conditions. Little conflict would be anticipated in both conditions due to 1) distinctive visuospatial cues in each arm that would be predictive of WBS delivery for one mouse but would have no predictive value for the other mouse and 2) the lack of material reward stimuli (sight, odor of food reward) that might promote competition.

II. Conditioning interpretation of heightened conflict/aggression with food reward

In contrast to WBS reward, food reward provides multiple distinctive cues (sight, odor, and location of food) for appetitive and consummatory approach behavior, competition, and even an opportunity for obtaining reward via stealing. Even in the absence of successfully stealing food reward, exposure to these cues should be sufficient to promote responses to

the incorrect arm, competition in the food-delivery area, and aggression.

III. Food reward versus WBS reward magnitude/value

Differential reinforcement magnitude/intensity is an additional factor that makes interpretation of the results of this paper difficult. Because WBS can be extremely reinforcing – rodents will often choose hypothalamic electrical stimulation over food or water reinforcement – comparisons of learning rates for WBS versus food reinforcement learning curves can be difficult to interpret without unless reward magnitude/value is equated. This is a potential problematic factor in the current study where mice learned the task much faster with WBS than with food. Without attempting to equate reward value, it is difficult to determine whether the slower learning rate in the food procedure is due to social learning factors, differences in reward value (see especially Figure 1c), or perhaps experiencing more punishment during training (loud noise). I am less concerned about the problem of differential reward magnitude in this particular paper because the comparison between WBS and food reward is not the central issue, but reward value as a variable deserves consideration in the design of future studies where appropriate.

IV. Problems with sample size when comparing groups and other statistical concerns

The small number of pairs run in the food reward condition makes some comparisons between those conditions difficult to interpret, for example in payoff equity (see Supplementary Figure 3). Here again, I am less concerned about this problem in this particular paper because the comparison between WBS and food reward is not the central issue, but sample size should be a consideration in the design of future studies where appropriate.

V. Missing phenomenon – Rapid rule transfer versus gradual acquisition

Finally, one of the most interesting manipulations in my opinion was pair re-organization (Figure 3i). One feature of rule use is rapid rule transfer that is similar to rapid “insight.” Insight is characterized by a rapid transition from the state of “not knowing” to the state of “knowing”, often characterized as an “aha” moment. Similarly, rule transfer is characterized by immediate generalization of a rule from one problem to a new but similar problem. Unfortunately, as the data in Figure 3i shows, in the pair reorganization test, mice that were shifted to a new pair with the opposite contingency from training showed a standard acquisition curve rather than a dramatic immediate shift that would suggest rapid rule transfer. This outcome suggests that original learning may have been based on discrimination learning, though it is possible that even if rule learning was the basis of original learning in the task, the transfer may have been too difficult to allow for rapid rule transfer.

Reviewer #2 (Remarks to the Author):

In this study Choe et al. developed a new task to study the behavior of mice competing for a limited resource. Importantly this resource, either food rewards or wireless stimulation of medial forebrain bundle (WBS), could be obtained at two different locations in the maze (reward location was signaled by a cue light). The existence of two reward locations allows for a commonly seen conflict management solution observed in the wild, where animals define territories, such that they 'claim' ownership over resources within their territory and avoid competition in the territories others' (bourgeois strategy). Mice quickly learned to go to a start location to initiate a trial and collect rewards on the side of the maze indicated by a light cue. Once trained, mice were placed in the same maze in pairs and thus had to compete for the rewards. They found that mice dyads, competing for WBS, adopt a rule similar to the bourgeois strategy, avoiding direct aggressive conflict, increasing reward consumption and achieving high equity in amount of rewards received by the two animals in the pair. In essence, mice quickly come to occupy different sides of the maze such that one mouse collects rewards on one side and the other mouse collects rewards on the other side. They further show that the behavior of mice is not habitual, as swapping mouse pairs temporarily affects performance that is quickly re-established. Finally, the authors show that mice that comply with the territory-based rule reciprocate after the other mouse also complied. In contrast, they show tolerance rather than negative reciprocity in response to rule violation by the other mouse.

This is a very interesting study that shows that mice can adopt socially competent rules in a competitive context paving the way for a new avenue of research on mechanisms of social behavior. The simplicity of the task, together with the degree of automated control over its variables and the richness of the data it yields, makes this task ideal for the difficult pursuit of examining the underpinnings of social interactions.

Despite its interest, relevance and timeliness, I have a few concerns that I would like to see addressed.

Major Concern

I believe the authors do not fully explore the richness of the dataset at hand. A more detailed description of the behavior would be very informative. Some of the analysis I mention below are central to the authors' claims, others I believe would add to the manuscript but are not essential for establishing the claims made by the authors.

Analysis central to the major findings:

1) Most of the analysis of the behavior during conflict testing relies on a rule violation measure: the number of times a mouse collected reward from the other's territory (by arriving there first) or interrupted reward consumption by the mouse already in the reward area. Rule observance corresponds to trials where neither violation is performed, i.e. trials where the mouse collected reward in his own territory and is left undisturbed. The two forms of rule violation are never reported separately which is an issue. Do they evolve similarly across trials and sessions? Is there a difference in relative frequency of both forms of violation across Mobs/Mobs, Mobs/Mvio, Mvio/Mvio pairs?

I suspect learning about the two different violations might be different. Do both mice go initially to the same arm and thus get little reward? Which form of violation decreases first?

Given that preoccupation violations require a clear notion of territory, they may be more prone to errors and thus be higher initially decreasing more steeply with training? Analyzing the evolution of the two forms of violation will provide insight into how mice are learning rule observance.

2) As mentioned above, preoccupation violation implies a clear definition of territory. However, territories are not defined by design in the experiment, but emerge from the dyads' behavior. Hence, it is crucial a detailed analysis of territory formation along side the evolution of rule violations. This is especially true for the first few sessions, which is when mice are acquiring their rule observant behavior.

The authors analyze the evolution of territories over sessions, by counting the number of trials the 'left mouse' and the 'right mouse' chose the left or the right reward arms (sup fig S4). This analysis is however not very clear. If I understand correctly, on each trial the mice have 3 options, to go into the left arm, the right arm or stay in the middle. The graphs in fig 2b fig. S4 seem to show that initially mice don't finish many trials successfully, but already show some degree of segregation differently occupying the left or right arm.

i) Graphs on fig S4 should be clarified, are the counts corresponding to trials where one or the other mouse entered each of the reward arms? Does it include when both mice went to the same arm? Are the counts for each time the mice paid a visit to the reward arms, regardless of trial structure? I also believe that it would be better to plot the data taking the behavior of pairs into account, instead of showing averaged individual behavior. Rather than showing average ML or MR visits to the left and right arms, the authors could for example show ratio or difference of left vs. right choices of mice in a pair (ML_{left}/MR_{left}, or ML_{left} - MR_{left}) and then average that across pairs.

ii) Given the importance of territory establishment, more detailed quantification should be devoted to characterizing how it takes place. What is going on when mice stay in the center zone? The example occupancy map in fig. 2 is nice. Still, I believe that showing how representative of the population it is and how it evolves over trials and sessions is crucial. Plotting the difference in occupancy time between the two mice, pixel by pixel for sessions at different stages of testing, would be very useful. One could for example plot green for pixels visited more by the green mouse and red for pixels visited more by the red mouse, and having color intensity indicate magnitude of difference (from dark red or green when almost exclusively one of the rats visited a pixel, to white, where both mice spend equal amounts of time a pixel). How does the absolute difference in space occupancy in the first sessions predict rule-observance learning?

3) Another issue is that the authors mention but do not quantify errors during conflict testing, i.e. how often for example ML goes to the left side when the light cue on the right went on? How do these errors relate to rule observance? Could it be that the pairs that are more rule-observant make more of these errors initially, such that they get more rewards than the pairs where both mice that go for the correct location and therefore compete?

4) The authors demonstrate positive reciprocity and tolerance by analyzing the 60% of trials where the rewarded side changes from that in the prior trial. In the remaining 40% of trials reward was signaled for one or other animal for several trials in a row making reward observance more costly to the animal whose side was not signaled. Further insight into rule

observance behavior may be found by quantifying the animals' capacity to withhold rule violations over multiple unrewarded trials.

Analysis that would improve the paper

1) Is learning different for familiar and unfamiliar pairs? An analysis of rule observance in time for the two kinds of pairs would be interesting.

2) Although body weight and aggression does not correlate with payoff equity, how does it relate to rule-observance learning and magnitude?

Other comments

1) The methods could be clearer.

i) How does the video based color ID gate WBS delivery?

ii) Why only 5 of the 11 pairs that reached criterion in the food task were used later?

iii) It is unclear whether stimulation is broken if a mouse leaves the reward zone and whether it is re-initiated upon re-entry.

A couple of experiments that I consider not to be required for publication but would be very interesting and would add to our understanding of rule observant behavior in a competitive context:

1) If one of the reward arms would be closed such that territory based rules would not solve the conflict, would mice learn to take turns in getting food? How does adoption of an alternating strategy correlate with adoption of territorial strategies? Addressing these questions may provide some measure as to what extent rule observance behavior is dependent upon the environment, and to what extent it depends upon some representation of the rule itself.

2) Many factors can explain the difference in behavior with food and WBS. One is the ability to perceive the reward delivery to the other. If for example a tone would be delivered every time WBS was being delivered (both during training and testing), the mice could perceive reward delivery to the other, as is the case with food pellets. How would this affect behavior? How would asymmetries in reward affect behavior?

Reviewer #3 (Remarks to the Author):

The authors have developed and empirically used a fascinating method of wireless brain stimulation in free-moving mice. When implanted into the medial forebrain bundle, it allows accessing and stimulating a reward center. The authors use such wireless brain stimulation (WBS) for conditioning / learning experiments to test whether mice are able to learn "rule observance" behavior when interacting with a conspecific in a potential conflict situation. Rule observance behavior has already been shown for animals. Here, the focus is whether mice can orderly resolve conflict without aggression. I do not know the literature very well

but assume that mice do solve a lot of conflicts through olfactory communication (via urine marks that contain various information on the individual producing the marks). Ignoring this means to ignore an important aspect of how territoriality and access to resources is communicated in mice, resulting in dominance interactions without visible aggression. Furthermore, I could not find any information on the sex of the experimental mice, despite the fact that aggression is fundamentally influenced by sex (as well as age and hormonal status).

I nevertheless applaud the authors for their device that opens the potential of testing hypotheses on learning or conditioning. I further fully agree that mice can learn strategies or rules (which by itself is not surprising or new). In the context of rule observance behavior as a strategy to solve conflict over access to a limited resource, however, I see fundamental problems with the approach taken. The WBS experiment is based on the idea that mice are in conflict over access to the reward through brain stimulation (WBS), and solve that conflict by learning rules to "orderly response conflict over limited rewards" (lines 82-83). In my opinion, the experiment presented does not fulfill the assumptions of such a hypothesis, and the results can be interpreted in another (simpler) way than a "rule observance behavior".

1. No data or evidence is provided that the reward received through WBS is perceived as a "limited resource", and that the mice learn to associate the presence of a conspecific as the "factor" causing limitation of the resource. The reward always stopped after 5 sec (WBS-award for 5 sec, see Figure 2). No data are given on reward duration in case of approach of the conspecific (disrupted, or WBS-sudden-stop). Even if WBS was stopped after 3 or 4 sec in the "disrupted" situation, evidence is missing

a) that mice can tell between a stimulus duration of 3 sec versus 5 sec;

b) that a shorter reward (3 sec) is perceived as a less rewarding stimulus than a 5 sec stimulation;

c) that the mice associated a shorter (= less rewarding) stimulus with the presence of a conspecific.

Given that conceptual problem, I conclude that the data presented do not allow interpreting the results in the above mentioned context.

2. The authors compare the WBS treatment with a "food" reward situation. I find the "food" reward situation interesting. In contrast to the WBS treatment it actually refers to a situation with a "limiting resource". However, it further differs from WBS in other aspects, making any comparison between the 2 treatments obsolete. In other words: the fact whether aggression occurred in one but not the other (WBS) situation does not allow to interpret the results as indicative of rule observance behavior.

a) In the "food" treatment, mice had been previously food deprived; hungry mice may generally behave differently (different motivation) than non food deprived conspecifics.

b) In the "food" treatment, food pellets can be monopolized through aggression (avoiding that the hungry individual will have to share it with a conspecific), and mice do react to such a situation by being aggressive, as shown in Figure 1e; being aggressive thus will improve access to the food, but no such consequences are achieved in the WBS situation (in the latter case, own behavior will NOT immediately result in improved – prolonged – reward); aggression thus makes sense in the "food" context (in evolutionary terms), but not in the

WBS situation. As a result, we expect predispositions influencing learning (species-specific predispositions for learning are well known).

c) The "food" treatment is expected to affect different (more specific) aspect of the behavior as the brain stimulation (which may be rather unspecific – not associated with a specific stimulus in the environment; in contrast, olfactory or visual perception of a food pellet is a rather specific stimulus). If this is correct, both treatments can hardly be compared in the context discussed by the authors.

d) Animals often trade-off feeding against predation risk, and therefore may be more cautious when approaching a location where they expect food. This may result in a somewhat slower speed (movement) when perceiving the cue, and explain the difference in a few seconds delay (appr 3-4 sec) in movement time to the reward zone, in comparison to the WBS treatment (see Figure 1d) – even independent of any intraspecific aggression.

3. I am not convinced that the "conflict resolution experiment" allows testing the hypothesis on rule-observance behavior. Some initial, even slight, side preference may result in the pattern observed in Figure 3a. Only the mouse that first enters the zone denoted as the reward zone will receive a reward (WBS). The conspecific (that was a bit slower in entering the zone) will not get a reward here (despite having seen and followed the cue!). Both may now also differ in urine marking behavior (which has not been analyzed). This may induce the second individual to search more actively and more widely (trial-and-error learning, after previous strategies did not resulted in a reward anymore). Such higher activity may allow the second individual to have a higher probability to first access the opposite zone (when the cue refers to the opposite zone for the first time) and will be rewarded – the above mentioned first individual will not be rewarded here, and "sorting" will begin since mice learn that a cue will only result in a reward at a specific location (zone). To test such an idea it is important to analyze what each mouse does during and after the first round of the modified experiment, and how this influences what it does afterwards.

Nature Communications revision for MS# NCOMMS-17-01018-T

Title: Mice in conflict show rule-observance behavior enhancing long-term benefit

Point-by-point responses to referees' comments:

We have tried to revise the manuscript based on the valuable comments. The positions of the revised text in response to the comments are indicated by 'Line #-#' in this response sheet and are highlighted in the text. In addition, inspired though not requested by the comments we have added substantial amount of new statements including new analysis results in the text, which are printed in red.

We deeply appreciate the reviewers for critical and constructive comments, which have helped us improve the quality of our manuscript tremendously.

Reviewer #1 (Remarks to the Author):

The very interesting set of experiments described in this manuscript examined the behavior of mice learning to perform a two-armed maze task for either food reward or wireless brain stimulation (WBS) reward. The claim is that mice learned to observe social rules that allowed them to avoid conflict for mutual benefit and reward payoff equity. Although this claim may be true, the observed behavior also appears to be amenable to a simpler conditioning explanation. Thus, although the experiment is provocative, I cannot recommend publication until the authors can provide stronger evidence discounting the more parsimonious conditioning interpretation. Another concern is the small number of mouse pairs in the food reward condition, which made it difficult to interpret observed differences in learning between WBS and food reward conditions.

I. Conditioning interpretation of conflict resolution performance with WBS reward

An alternative to rule-observance is the idea that mice responded as they did due to discrimination learning. From this perspective, mice in the WBS reward condition learned to respond on cue to move to the left or right arm of the maze where they received WBS. WBS mice reinforced for responding to the left (ML) were never reinforced in the right arm of the maze, whereas mice reinforced for responding to the right (MR) were never reinforced in the left arm of the maze. Visual cues were associated with the appropriate goal when the correct mouse was present, so mice could learn that flashing lights in one location signaled WBS, whereas flashing lights in the opposite arm, signaling no WBS, were irrelevant. Critically, MR mice could not compete with ML mice for WBS in the left arm because no behavior – competitive or otherwise – could ever lead to a reinforcer in the incorrect arm. Furthermore, because there were no physical reinforcers involved in WBS, mice could not even observe other mice receiving a physical reinforcer. Thus, responses by WBS mice to the incorrect arm should not increase above some baseline level. The opposite would be experienced by MR and ML mice in the opposite (Right) arm of the maze. The predicted results would be rapid learning of distinct responses for left and right arms for both WBS conditions. Little conflict would be anticipated in both conditions due to 1) distinctive visuospatial cues in each arm that would be predictive of WBS delivery for one mouse but would have no predictive value for the other mouse and 2) the lack of material reward stimuli (sight, odor of food reward) that might promote competition.

- ⇒ Upon this critical comment we realized that our presentation in the previous manuscript was not sufficient to give an accurate description of the behavior tests. To explain briefly, the mice are first operant-trained individually to start/seek the reward. When the mouse enters the start

zone a light cue comes on one of the two arms (randomly selected). Then, only if the mouse enters into the designated corner of this arm (correct choice) it gets the WBS reward. Only those mice that passed this operant conditioning have been selected and subjected to the main, pair test, in which a pair of mice will compete for the reward. Therefore each mouse of a pair has already learned that the light cue indicates that the reward will be given in the corner of that arm. In the main test, one can see that each mouse experiences to get reward from both arms in the earlier sessions. After several sessions a preferred side for each mouse in a pair becomes apparent. Fig 3a (revised MS) shows the representative plots of the results of the pair test, comparing the first quarter and the last quarter of the twenty sessions. We have also added in Fig S7 the plots of the results of all the individual sessions for each of the 19 pairs. The gradual development of the allocation behavior through the training sessions is apparent. These results would rule out the alternative explanation, discrimination learning, for the behavioral results.

- ⇒ It is true that a mouse may be unable to observe the other receiving WBS reward which is not a physical reinforcer. Nevertheless, judging from the training protocol and the behavioral results we can conclude that the mouse is aware that the partner, instead of itself, is receiving the reward on the visually cued arm.
- ⇒ We have revised Figure 3 and added Fig. S7 to give a better presentation of the result of the conflict resolution test. The text has been revised accordingly: Line150 ~ 171.

II. Conditioning interpretation of heightened conflict/aggression with food reward

In contrast to WBS reward, food reward provides multiple distinctive cues (sight, odor, and location of food) for appetitive and consummatory approach behavior, competition, and even an opportunity for obtaining reward via stealing. Even in the absence of successfully stealing food reward, exposure to these cues should be sufficient to promote responses to the incorrect arm, competition in the food-delivery area, and aggression.

- ⇒ We fully agree with the reviewer's explanation on the behavior of the mouse with food reward. With gratitude, we dealt with this issue in the discussion part of our manuscript. (Line 305 ~ 307)

III. Food reward versus WBS reward magnitude/value

Differential reinforcement magnitude/intensity is an additional factor that makes interpretation of the results of this paper difficult. Because WBS can be extremely reinforcing – rodents will often choose hypothalamic electrical stimulation over food or water reinforcement – comparisons of learning rates for WBS versus food reinforcement learning curves can be difficult to interpret without unless reward magnitude/value is equated. This is a potential problematic factor in the current study where mice learned the task much faster with WBS than with food. Without attempting to equate reward value, it is difficult to determine whether the slower learning rate in the food procedure is due to social learning factors, differences in reward value (see especially Figure 1c), or perhaps experiencing more punishment during training (loud noise). I am less concerned about the problem of differential reward magnitude in this particular paper because the comparison between WBS and food reward is not the

central issue, but reward value as a variable deserves consideration in the design of future studies where appropriate.

- ⇒ We agree with the reviewer's concern on whether WBS reward has equal value and intensity to food pellet. As the reviewer pointed out, our main goal in the current work is not to compare between WBS and food reward. Still, we thank the reviewer for suggestions which would be very important to the future direction, and we have mentioned this concern in our revised discussion. [Line 306 ~ 307]

IV. Problems with sample size when comparing groups and other statistical concerns

The small number of pairs run in the food reward condition makes some comparisons between those conditions difficult to interpret, for example in payoff equity (see Supplementary Figure 3). Here again, I am less concerned about this problem in this particular paper because the comparison between WBS and food reward is not the central issue, but sample size should be a consideration in the design of future studies where appropriate.

- ⇒ We totally agree with the reviewer's comment on the small sample size for the food reward group. We would like to thank the reviewer's generous remark that he/she is less concerned about this problem in this particular paper. Nevertheless we would like to carry out another batch of experiments for the food reward group, although we would not be able to include the additional data this time due to the long time it takes to complete our behavior test.

V. Missing phenomenon – Rapid rule transfer versus gradual acquisition

Finally, one of the most interesting manipulations in my opinion was pair re-organization (Figure 3i). One feature of rule use is rapid rule transfer that is similar to rapid “insight.” Insight is characterized by a rapid transition from the state of “not knowing” to the state of “knowing”, often characterized as an “aha” moment. Similarly, rule transfer is characterized by immediate generalization of a rule from one problem to a new but similar problem.

Unfortunately, as the data in Figure 3i shows, in the pair reorganization test, mice that were shifted to a new pair with the opposite contingency from training showed a standard acquisition curve rather than a dramatic immediate shift that would suggest rapid rule transfer. This outcome suggests that original learning may have been based on discrimination learning, though it is possible that even if rule learning was the basis of original learning in the task, the transfer may have been too difficult to allow for rapid rule transfer.

- ⇒ We are thankful to the reviewer for this absolutely important comment, the “aha moment.” We have expanded the horizontal axis of the figure configuration so that the early event after the pair re-organization can be easily visible. Fig 3i of the previous MS has become Fig 5 in the revised manuscript. It is now clearly visible that the learning curve steeply increases immediately after the pair re-organization, much faster compared to the standard acquisition curve. This dramatic, immediate shift of the learning curve would suggest a rapid rule transfer. We have also added statistical measures to show the clear and significant difference between the two curves. Thus, now we are more confident that the mice have developed rule-observance behavior that differs from simple reinforcement learning. This point has been integrated in the revised manuscript. (Lines 229 ~ 235)

Reviewer #2 (Remarks to the Author):

In this study Choe et al. developed a new task to study the behavior of mice competing for a limited resource. Importantly this resource, either food rewards or wireless stimulation of medial forebrain bundle (WBS), could be obtained at two different locations in the maze (reward location was signaled by a cue light). The existence of two reward locations allows for a commonly seen conflict management solution observed in the wild, where animals define territories, such that they 'claim' ownership over resources within their territory and avoid competition in the territories others' (bourgeois strategy). Mice quickly learned to go to a start location to initiate a trial and collect rewards on the side of the maze indicated by a light cue. Once trained, mice were placed in the same maze in pairs and thus had to compete for the rewards. They found that mice dyads, competing for WBS, adopt a rule similar to the bourgeois strategy, avoiding direct aggressive conflict, increasing reward consumption and achieving high equity in amount of rewards received by the two animals in the pair. In essence, mice quickly come to occupy different sides of the maze such that one mouse collects rewards on one side and the other mouse collects rewards on the other side. They further show that the behavior of mice is not habitual, as swapping mouse pairs temporarily affects performance that is quickly re-established. Finally, the authors show that mice that comply with the territory-based rule reciprocate after the other mouse also complied. In contrast, they show tolerance rather than negative reciprocity in response to rule violation by the other mouse.

This is a very interesting study that shows that mice can adopt socially competent rules in a competitive context paving the way for a new avenue of research on mechanisms of social behavior. The simplicity of the task, together with the degree of automated control over its variables and the richness of the data it yields, makes this task ideal for the difficult pursuit of examining the underpinnings of social interactions.

Despite its interest, relevance and timeliness, I have a few concerns that I would like to see addressed.

Major Concern

I believe the authors do not fully explore the richness of the dataset at hand. A more detailed description of the behavior would be very informative. Some of the analysis I mention below are central to the authors' claims, others I believe would add to the manuscript but are not essential for establishing the claims made by the authors.

Analysis central to the major findings:

1) Most of the analysis of the behavior during conflict testing relies on a rule violation measure: the number of times a mouse collected reward from the other's territory (by arriving there first) or interrupted reward consumption by the mouse already in the reward area. Rule observance corresponds to trials where neither violation is performed, i.e. trials where the mouse collected reward in his own territory and is left undisturbed. The two forms of rule violation are never reported separately which is an issue. Do they evolve similarly across trials and sessions? Is there a difference

in relative frequency of both forms of violation across Mobs/Mobs, Mobs/Mvio, Mvio/Mvio pairs? I suspect learning about the two different violations might be different. Do both mice go initially to the same arm and thus get little reward? Which form of violation decreases first? Given that preoccupation violations require a clear notion of territory, they may be more prone to errors and thus be higher initially decreasing more steeply with training? Analyzing the evolution of the two forms of violation will provide insight into how mice are learning rule observance.

⇒ Being stimulated by the reviewer's comment that we have not provided detailed information on the rule violation behavior, we have revised the manuscript including presentation of the evolution of the two forms of rule violation. In doing this revision we are quite impressed by the tremendous insights and predictions offered by the reviewer regarding the issue. We can find from the graphs (Fig 4h) that pre-emptive occupation decreases earlier and efficiently in the M_{obs} . This may be due to the significant amount of efforts that is needed for a successful pre-emptive occupation of the partner's reward side. On the other hand, the proportion of disruption gradually decreases as the number of sessions with rule observance increases, although the actual number of disruption did not reduce much through the sessions. These may suggest that the major way the mice learn rule observance is through increasing the actual number of sessions without violations. It appears possible that they may still retain, but suppress, the urge to disrupt the partner's reward. We have revised the figures and text incorporating these points. Line#220 ~ 224.

2) As mentioned above, preoccupation violation implies a clear definition of territory. However, territories are not defined by design in the experiment, but emerge from the dyads' behavior. Hence, it is crucial a detailed analysis of territory formation along side the evolution of rule violations. This is especially true for the first few sessions, which is when mice are acquiring their rule observant behavior.

The authors analyze the evolution of territories over sessions, by counting the number of trials the 'left mouse' and the 'right mouse' chose the left or the right reward arms (sup Fig S4). This analysis is however not very clear. If I understand correctly, on each trial the mice have 3 options, to go into the left arm, the right arm or stay in the middle. The graphs in Fig 2b Fig. S4 seem to show that initially mice don't finish many trials successfully, but already show some degree of segregation differently occupying the left or right arm.

i) Graphs on Fig S4 should be clarified, are the counts corresponding to trials where one or the other mouse entered each of the reward arms? Does it include when both mice went to the same arm? Are the counts for each time the mice paid a visit to the reward arms, regardless of trial structure? I also believe that it would be better to plot the data taking the behavior of pairs into account, instead of showing averaged individual behavior. Rather than showing average ML or MR visits to the left and right arms, the authors could for example show ratio or difference of left vs. right choices of mice in a pair (MLleft/MRleft, or MLleft - MRleft) and then average that across pairs.

⇒ In previous data, all counts were only for the rewarded mice. The count does not include when both mice went to the same arm. To clarify many questions on the counts, we adopted the reviewer's suggestion to plot the average of the difference of left vs. right choices (MLleft-MRleft & MRright-MLright). This result displayed in Fig 3c confirms that the average tendency was corresponding to the representative data. We thank the reviewer for this suggestion which helped improve our paper. [lines #167 ~ 170]

ii) Given the importance of territory establishment, more detailed quantification should be devoted to characterizing how it takes place. What is going on when mice stay in the center zone?

Left cue		WBS-ON			WBS-OFF		
Pair type	Pair#	correct arm	center zone	incorrect arm	correct arm	center zone	incorrect arm
Mobs-Mobs	1	1.0	88.9	10.1	15.6	56.7	27.8
	2	2.0	88.8	9.2	8.7	60.9	30.4
	3	1.0	86.0	13.0	13.5	71.9	14.6
	4	2.0	92.0	6.0	24.6	40.6	34.8
	5	1.0	82.3	16.7	21.0	27.2	51.9
	6	5.1	92.9	2.0	22.5	48.8	28.8
	7	2.7	68.9	28.4	9.2	63.1	27.7
	8	1.0	88.0	11.0	16.1	24.2	59.7
Average		2.0	86.0	12.0	16.4	49.2	34.4
SEM		0.5	2.5	2.6	2.0	5.7	4.8
Mvio-Mvio	16	4.2	74.0	21.9	19.7	57.7	22.5
	17	75.6	11.6	12.8	75.8	18.2	6.1
	18	72.1	18.6	9.3	79.2	12.5	8.3
	19	73.7	18.4	7.9	86.8	5.3	7.9
Average		56.4	30.7	13.0	65.4	23.4	11.2
SEM		15.1	12.6	2.7	13.3	10.2	3.3

⇒

⇒ In order to get an idea how the territory establishment is achieved in $M_{obs}-M_{obs}$ pairs compared to $M_{vio}-M_{vio}$ pairs, we analyzed the position of the opponent mouse at the time points when WBS reward delivery was initiated or terminated, respectively, during the last five sessions. We compared $M_{obs}-M_{obs}$ pairs and the $M_{vio}-M_{vio}$ pairs in the proportions of trials where the mice are positioned among the three areas, correct arm, center zone, or the incorrect arm. In $M_{obs}-M_{obs}$ pairs, at the moment when WBS reward was initiated by one mouse, the majority of the opponent remained in the center zone ($86.0 \pm 2.5\%$) and very few in the correct arm ($2 \pm 0.5\%$). After five seconds of WBS reward, large proportion of mice who were staying in the center area moved away. Interestingly, majority of them moved into the incorrect arm, in effect to stay away from the correct arm where the partner receives the reward. In contrast, in the $M_{vio}-M_{vio}$ pairs, even at the reward initiation time, majority of the opponent were in the correct arm ($56.4 \pm 15.1\%$). At the reward termination time, even more mice were positioned in the correct arm ($65.4 \pm 13.3\%$). Therefore, these results imply that the rule-observant mice show active effort not to disrupt the partner's reward. [Supplementary table. 2, Discussion line #273 ~ 287]

The example occupancy map in fig. 2 is nice. Still, I believe that showing how representative of the population it is and how it evolves over trials and sessions is crucial. Plotting the difference in occupancy time between the two mice, pixel by pixel for sessions at different stages of testing, would be very useful. One could for example plot green for pixels visited more by the green mouse and red for pixels visited more by the red mouse, and having color intensity indicate magnitude of difference (from dark red or green when almost exclusively one of the rats visited a pixel, to white, where both mice spend equal amounts of time a pixel) How does the absolute difference in space occupancy in the first sessions predict rule-observance learning?

⇒ We found the reviewer's suggestion of pixel-based presentation a wonderful idea to show our result very efficiently. We have plotted pixel-by-pixel occupancy change of every 19 pairs in Fig 3a. In addition to this figure, we have also included the graphics showing position of each

mouse for all the sessions of all the pairs, a kind of raw data, in Fig S7. Revised text is added in the manuscript. [lines #150~159]

	1 st session		20 th session	
	Z _L	Z _R	Z _L	Z _R
Pair1	0.50	0.56	1.00	1.00
Pair2	0.75	1.00	1.00	0.90
Pair3	0.09	0.27	0.90	1.00
Pair4	0.82	0.60	1.00	1.00
Pair5	0.20	1.00	1.00	1.00
Pair6	0.09	-0.45	0.56	0.88
Pair7	-0.71	0.43	1.00	1.00
Pair8	-0.50	-0.14	1.00	1.00
Pair9	0.43	0.71	0.56	1.00
Pair10	0.27	1.00	1.00	1.00
Pair11	0.20	0.00	1.00	1.00
Pair12	1.00	1.00	1.00	1.00
Pair13	-1.00	0.00	0.89	1.00
Pair14	-0.60	0.67	0.87	1.00
Pair15	0.60	0.14	1.00	1.00
Pair16	-0.09	-0.64	0.90	0.90
Pair17	-0.50	0.43	0.00	0.17
Pair18	0.25	0.56	-0.25	-0.71
Pair19	0.45	1.00	-0.40	-0.40

⇒ We calculated the absolute difference in space occupancy for each zone in the first session (For example, M_L occupancy rate for left reward zone – M_R occupancy rate for left reward zone, vice versa for the right side) of 19pairs. Positive values indicate that M_L occupied the left zone more frequently than M_R . To examine whether it can predict rule-observance learning, we divided the table according to the three pair types($M_{obs}-M_{obs}$, $M_{obs}-M_{vio}$, $M_{vio}-M_{vio}$).For all kinds of pair types, we observed wide range of the absolute difference value between -1 and +1. There was no significant pattern associated with the pair types. And even among $M_{obs}-M_{obs}$ pairs, three out of eight pairs exhibited change of preferring reward arm eventually. These results show that rule-observance behavior cannot be predicted in the first session.

3) Another issue is that the authors mention but do not quantify errors during conflict testing, i.e. how often for example ML goes to the left side when the light cue on the right went on? How do these errors relate to rule observance? Could it be that the pairs that are more rule-observant make more of these errors initially, such that they get more rewards than the pairs where both mice that go for the correct location and therefore compete?

⇒ Please check our supplementary Figure 7. The case the reviewer mentioned as “error” surely happens frequently. However, as we counted every error trials, there was no significant pattern of difference depending on pair types or sessions. Therefore, we can say that the proportion the error takes in early sessions could not be a major cause to accelerate rule-observance.

4) The authors demonstrate positive reciprocity and tolerance by analyzing the 60% of trials where the rewarded side changes from that in the prior trial. In the remaining 40% of trials reward was signaled for one or other animal for several trials in a row making reward observance more costly to the animal whose side was not signaled. Further insight into rule observance behavior may be found by quantifying the animals' capacity to withhold rule violations over multiple unrewarded trials.

⇒ It is expected that repeated reward to the opponent mouse would evoke stress for a mouse to show stable positive reciprocity. We analyzed the remaining 40% of trials where the rewarded side was not changed. About 26% was own-other's-other's turn, and 11% was own-other's-other's-other's turn. Although the number of case is low, we analyzed the conditional probabilities of positive reciprocity for each group, M_{obs} and M_{vio} . M_{obs} showed high level of steady rule-observance than the M_{vio} (Fig. 6c). It indicates that rule-observance behavior is very strong and stable for those mice. This issue is included in the Fig 6 and the text on our manuscript. [line #252 ~257, #298 ~ 304]

Analysis that would improve the paper

1) Is learning different for familiar and unfamiliar pairs? An analysis of rule observance in time for the two kinds of pairs would be interesting.

⇒ When we designed the experiments we did not consider the familiarity between members in pairs. Upon examination of the record, we see that the dyads are mostly strangers to each other. Therefore we cannot answer this question with current results. As the reviewer mentioned, we find the familiarity factor can be very interesting to investigate, and it will be an issue to be included in our future study. We have mentioned this issue in the discussion [line 316 ~ 319].

2) Although body weight and aggression does not correlate with payoff equity, how does it relate to rule-observance learning and magnitude?

⇒ There was almost no aggressive behavior within the WBS group, and thus we cannot say about the correlation between aggression and rule-observance. For the body weight, we could not see neither positive nor negative correlation to the rule-observance learning and magnitude. We have included the plots regarding this issue in our supplementary figure 6, and mentioned in the text [line #224 ~ 228].

Other comments

1) The methods could be clearer.

i) How does the video based color ID gate WBS delivery?

⇒ WBS delivery is not based on color ID. It is based on the location within the arena. IR signal was delivered into the reward zones. Therefore IR signal delivered to each reward zone triggers the electronic circuit on the headset, and generates the current for deep brain stimulation.

ii) Why only 5 of the 11 pairs that reached criterion in the food task were used later?

⇒ The number 11 was not indicating pairs but the number of individual mice. Since we had 11 well-trained mice that passed the conditioning, we could pair them into 5 pairs and used them for further experiments.

iii) It is unclear whether stimulation is broken if a mouse leaves the reward zone and whether it is re-initiated upon re-entry.

⇒ The stimulation occurs only when the IR signal is delivered to the receiver on the headset. Naturally, stimulation will be cut off when the mouse leaves the reward zone which is shed with IR signal. However, if the mouse re-enters the reward zone, it will get the stimulation again for the remaining reward duration as long as not disrupted by the partner. We included this explanation into our methods description. [supplementary methods lines #61 ~ 62]

A couple of experiments that I consider not to be required for publication but would be very interesting and would add to our understanding of rule observant behavior in a competitive context:

1) If one of the reward arms would be closed such that territory based rules would not solve the conflict, would mice learn to take turns in getting food? How does adoption of an alternating strategy correlate with adoption of territorial strategies? Addressing these questions may provide some measure as to what extent rule observance behavior is dependent upon the environment, and to what extent it depends upon some representation of the rule itself.

This is a great suggestion. We can think of several variations of this wonderful idea. Will the rule observance behavior still prevails when there is only one physical source of reward? This test may be carried out with the rearranged pairs chosen from the M_{obs} - M_{obs} pairs. Thank you for such a great idea.

2) Many factors can explain the difference in behavior with food and WBS. One is the ability to perceive the reward delivery to the other. If for example a tone would be delivered every time WBS was being delivered (both during training and testing), the mice could perceive reward delivery to the other, as is the case with food pellets. How would this affect behavior? How would asymmetries in reward affect behavior?

⇒ We thank this reviewer for the insightful comments on our conflict resolution test. We agree with the idea that coupling a marker to the WBS reward will help mice to get more information on the reward. In our current study, we gave aversive tone as a punishment for incorrect choice. Thus, if we give non-aversive tone coupled with the brain stimulation, mice would know better on the properties of WBS, such as the amount of the reward they actually received. In addition, the follower mice would also know better whether their partners are getting reward instead of them. We are curious whether such modification will facilitate or

delay the emergence of rule observance behavior.

- ⇒ The effect of asymmetries in reward on the rule-observance behavior is in fact what we have started to test for the follow up study.
- ⇒ We thank the reviewer for many great suggestions which will be very important in the future direction of experiments.

Reviewer #3 (Remarks to the Author):

The authors have developed and empirically used a fascinating method of wireless brain stimulation in free-moving mice. When implanted into the medial forebrain bundle, it allows accessing and stimulating a reward center. The authors use such wireless brain stimulation (WBS) for conditioning / learning experiments to test whether mice are able to learn “rule observance” behavior when interacting with a conspecific in a potential conflict situation. Rule observance behavior has already been shown for animals. Here, the focus is whether mice can orderly resolve conflict without aggression.

I do not know the literature very well but assume that mice do solve a lot of conflicts through olfactory communication (via urine marks that contain various information on the individual producing the marks). Ignoring this means to ignore an important aspect of how territoriality and access to resources is communicated in mice, resulting in dominance interactions without visible aggression.

- ⇒ Olfactory communication and territoriality by urine marking have been reported in various kinds of animal models. We think the reviewer’s concern to be very reasonable. Practically, we have observed the frequency of excretion was high in the initial stages and decreases as the sessions go by. Unfortunately, we could not analyze this issue due to the difficult problem of distinguishing which urine marks belong to which mouse. As we found that mice are capable of resolving conflict by reward zone allocation, olfactory communication might be one of the methods for a mouse to make a choice of its preferred side and let the partner be aware of its decision. We mentioned this precious comment in the discussion part of our manuscript for the moment, and we would be eager to test on urine marking as we find a proper tool. [Line310 ~ 314]

Furthermore, I could not find any information on the sex of the experimental mice, despite the fact that aggression is fundamentally influenced by sex (as well as age and hormonal status).

- ⇒ Sex and age were written in the supplementary methods.
“Male C57BL/6J mice were used for the current study. Four or five mice were housed in a cage under a 12:12 light-dark cycle until the 11th week in age. During the time, food and water were accessible ad libitum.” (Supplementary Methods lines 4 ~ 6)

I nevertheless applaud the authors for their device that opens the potential of testing hypotheses on learning or conditioning. I further fully agree that mice can learn strategies or rules (which by itself is not surprising or new). In the context of rule observance behavior as a strategy to solve conflict over

access to a limited resource, however, I see fundamental problems with the approach taken. The WBS experiment is based on the idea that mice are in conflict over access to the reward through brain stimulation (WBS), and solve that conflict by learning rules to “orderly response conflict over limited rewards” (lines 82-83). In my opinion, the experiment presented does not fulfill the assumptions of such a hypothesis, and the results can be interpreted in another (simpler) way than a “rule observance behavior”.

1. No data or evidence is provided that the reward received through WBS is perceived as a “limited resource”, and that the mice learn to associate the presence of a conspecific as the “factor” causing limitation of the resource. The reward always stopped after 5 sec (WBS-award for 5 sec, see Figure 2). No data are given on reward duration in case of approach of the conspecific (disrupted, or WBS-sudden-stop). Even if WBS was stopped after 3 or 4 sec in the “disrupted” situation, evidence is missing

a) that mice can tell between a stimulus duration of 3 sec versus 5 sec;

b) that a shorter reward (3 sec) is perceived as a less rewarding stimulus than a 5 sec stimulation;

n=10	6sec		2sec		%visit6sec	
Day	mean	SE	mean	SE		
1	5.1	0.887568	6.4	0.921352	0.54	0.058564
2	7.9	1.779201	8.3	1.686877	0.64	0.073879
3	9.5	1.808928	8	1.498147	0.67	0.042881
4	9.9	1.386042	6.9	1.16857	0.64	0.073464
5	12.1	1.93477	7.3	1.813836	0.76	0.052988
6	14.3	2.011357	5.7	2.011357	0.85	0.045

n=10	6sec		4sec		%visit6sec	
Day	mean	SE	mean	SE		
1	7.4	0.968389	8.9	1.17804	0.45	0.051474
2	9.4	1.752459	9.4	1.661325	0.50	0.086968
3	12.9	1.797838	7.1	1.797838	0.65	0.089892
4	12.4	2.181742	7.4	2.082733	0.62	0.108432
5	13.7	1.751507	6.3	1.751507	0.69	0.087575
6	13.9	1.888268	6.1	1.888268	0.70	0.094413

⇒ Unlike there is freezing behavior for fear response, we do not have simple behavioral markers to use in quantifying how much the mice are rewarded. We can only tell whether mice prefer a longer reward to a shorter reward by comparing the time they visit the two reward zones with different duration of IR lighting time. In the experiments where we try to establish the stimulation reward protocol, we found that mice chose the longer over shorter stimulation reward: 6sec over 2sec, or 6sec over 4sec, conforming to ‘the matching rule’. We could interpret these results that they can distinguish the different durations of the stimulus, and thus perceive them as different amounts of reward.(table above) We have included this point in the method. [Supplementary Method line#67 ~ 74, Supplementary table 1]

c) that the mice associated a shorter (= less rewarding) stimulus with the presence of a conspecific.

- ⇒ We understand that this is a very critical issue. To address this issue, we have carried out control experiments, the results of which are added as Figure S4 in the revised manuscript. In order to address this issue directly, we have modified the protocol where the WBS reward is not cut off even when the opponent mouse has entered the reward zone (the non-disruption protocol). When the WBS reward is delivered for 5 seconds regardless of the presence of the other, the behavior pattern of the pair are quite different from that of the main experiments (the disruption protocol). 1). Both of the mice in a pair run toward the light cued reward zone as soon as the light cue is on. The difference in the behavior pattern between the two protocols is the most dramatic at the reward termination especially in the later sessions: essentially all trials end with the two mice within the reward zone. 2). The pairs reach the maximum trial number per session (40 trials within 20 minutes) much faster. Their behavior is essentially the same as those during the first conditioning step, the reward start/seeking test. They learned that the presence of a partner mouse does not bring any difference to their chance of getting the maximum reward. All they need to do is to start the reward cycle as many times as possible by co-entering the start zone. These dramatic differences in the animals' behavior between the two protocols suggest that under the disruption protocol both of the mice, the one receiving the reward in its zone and the opponent not disrupting the other's reward, perceive that disrupting reward will result in less than the maximum amount of the reward they could obtain. Such perception would have driven them to establish the reward allocation behavior, which then enhanced their reward amount in a mutually beneficial way.
- ⇒ Figure S4, added. The results and interpretation are described in (lines 178 ~ 192).

Given that conceptual problem, I conclude that the data presented do not allow interpreting the results in the above mentioned context.

- ⇒ We hope that the results of our additional control experiments and our interpretation could satisfy the reviewer regarding this issue.

2. The authors compare the WBS treatment with a “food” reward situation. I find the “food” reward situation interesting. In contrast to the WBS treatment it actually refers to a situation with a “limiting resource”. However, it further differs from WBS in other aspects, making any comparison between the 2 treatments obsolete. In other words: the fact whether aggression occurred in one but not the other (WBS) situation does not allow to interpret the results as indicative of rule observance behavior.

- ⇒ First, we would like to emphasize that it is not our major issue to compare the effect of WBS over food reward. We found WBS works as a positive reinforcement compared to sham-WBS in Fig. 1d. We performed the experiments with food reward as a standard positive reinforcement since we were trying new reinforcement tools. Unexpectedly, we could observe that under the WBS conditions mice displayed interesting behavior, which has led us to study further.

a) In the “food” treatment, mice had been previously food deprived; hungry mice may generally behave differently (different motivation) than non food deprived conspecifics.

- ⇒ We agree with the reviewer's description about different behaviors between the food-deprived and non-food-deprived mice.

b) In the “food” treatment, food pellets can be monopolized through aggression (avoiding that the hungry individual will have to share it with a conspecific), and mice do react to such a situation by being aggressive, as shown in Figure 1e; being aggressive thus will improve access to the food, but no such consequences are achieved in the WBS situation (in the latter case, own behavior will NOT immediately result in improved – prolonged – reward); aggression thus makes sense in the “food” context (in evolutionary terms), but not in the WBS situation. As a result, we expect predispositions influencing learning (species-specific predispositions for learning are well known).

⇒ We agree with the reviewer’s explanations regarding the behavioral differences between the food reward vs. WBS. We are also curious how the mouse will respond when the rest amount of reward is STOLEN by the other that interrupted its own WBS delivery. We would be very excited to test this by experiment, but our current system cannot give IR signal selectively to one mouse over the other: infrared signal is given by the distinct area within the maze. With gratitude, we dealt with this concern in the discussion part of our manuscript. [Lines 307 ~ 310]

c) The “food” treatment is expected to affect different (more specific) aspect of the behavior as the brain stimulation (which may be rather unspecific – not associated with a specific stimulus in the environment; in contrast, olfactory or visual perception of a food pellet is a rather specific stimulus). If this is correct, both treatments can hardly be compared in the context discussed by the authors.

⇒ We understood that our comparison between food reward and WBS could be inappropriate as you mentioned. We would like to say, however, that this comparison was not the main issue we wanted to study in the current work.

d) Animals often trade-off feeding against predation risk, and therefore may be more cautious when approaching a location where they expect food. This may result in a somewhat slower speed (movement) when perceiving the cue, and explain the difference in a few seconds delay (appr 3-4 sec) in movement time to the reward zone, in comparison to the WBS treatment (see Figure 1d) – even independent of any intraspecific aggression.

⇒ We agree with the reviewer’s comment on the possibility that the trade-off feeding against predation risk is causing the difference in speed in perceiving the cue.

3. I am not convinced that the “conflict resolution experiment” allows testing the hypothesis on rule-observance behavior. Some initial, even slight, side preference may result in the pattern observed in Figure 3a. Only the mouse that first enters the zone denoted as the reward zone will receive a reward (WBS). The conspecific (that was a bit slower in entering the zone) will not get a reward here (despite having seen and followed the cue!). Both may now also differ in urine marking behavior (which has not been analyzed). This may induce the second individual to search more actively and more widely (trial-and-error learning, after previous strategies did not resulted in a reward anymore). Such higher activity may allow the second individual to have a higher probability to first access the opposite zone (when the cue refers to the opposite zone for the first time) and will be rewarded – the above mentioned first individual will not be rewarded here, and “sorting” will begin since mice learn that a cue will only result in a reward at a specific location (zone). To test such an idea it is important to analyze what each mouse does during and after the first round of the modified experiment, and how this influences what it does afterwards.

- ⇒ We thank the reviewer for in-depth discussion and an important suggestion. Following the suggestion we have closely analyzed the behavior of the mice during and after the first round of conflict resolution experiment. In addition we have added in Fig. S7 the plots of the results of all the individual sessions for each of the 19 pairs. The analysis reveals that 1) The reward direction of the first trial of the first session does not predict the final direction of the reward zone allocation. See table below. 2).During the first session (40 trials) of the conflict resolution experiment both mice of a pair experience reward from both of the arm although the ratio is not 1:1.3) The gradual development of the allocation behavior through the 20 training sessions is apparent.
- ⇒ Perhaps, the most important finding that support our interpretation of rule-observance behavior may be the results in Fig. 5, showing the rapid re-establishment of the new reward zone allocation (rapid rule transfer) when the pairs were re-organized with the same-side-preferring mice.
- ⇒ We analyzed which mouse was rewarded in first trials for each reward zone, then checked if it changes or sustains enduring the sessions. In the majority of all the pairs, a dominant one gets rewarded in both sides initially, but gradually gives up one of the two arms rather than trying to keep both reward zones. Among seven pairs that exhibit split behavior in the beginning, four of them showed same space occupancy at the last session. In the remaining three pairs, the rewarding arm was eventually changed. Important point to note is that, in no cases, the initial place occupancy by one mouse is sustained all the way to the final session without changing during the process. These results shows that the most plausible explanation of our data appears that the competing mice developed rule observance behavior to enhance their mutual benefit under the experimental conditions described.

Type	pair#	First trial of first session		Final result		Both arms	Split		Total
		L	R	L	R		Sustained	Changed	
Mobs vs Mobs	1	R	R	R	G	3	3	2	8
	2	R	G	R	G				
	3	G	G	R	G				
	4	G	R	G	R				
	5	G	G	R	G				
	6	R	G	G	R				
	7	R	G	R	G				
	8	G	R	R	G				
Mobs vs Mvio	9	G	R	G	R	6	1	0	7
	10	G	G	R	G				
	11	G	G	G	R				
	12	G	G	R	G				
	13	R	R	G	R				
	14	G	G	G	R				
Mvio vs Mvio	15	G	G	G	R	3	0	1	4
	16	G	G	R	G				
	17	G	G	R	G				
	18	R	R	R	G				
	19	G	R	R	G				
Total						12	4	3	19

- ⇒ Black-lettered pairs indicate those where one dominant mouse monopolized first trials of both arms. Red-texted pairs indicate those exhibited split behavior in the first trial of first session. Green-texted pairs are those where the rewarding arm in the final session was different from that in the first trial of the first session.

Editorial Note: Note that there is no second review provided by Reviewer 3. This is because this reviewer was unable to provide a timely, re-review of the manuscript. In such cases, we ask the other reviewers to indicate whether or not the revised manuscript addressed the concerns raised by the absent reviewer, and then decide editorially whether or not to find a new reviewer. In this case, both we and the other two reviewers felt that the authors had addressed all the concerns.

REVIEWERS' COMMENTS:

Reviewer #1 (Remarks to the Author):

Review of Manuscript Number: NCOMMS-17-01018A

Title: Mice in conflict show rule-observance behavior enhancing long-term benefit

The authors have made a great effort to address all concerns raised by the reviewers. With regard to criticisms raised by this reviewer, the authors were not able to address all concerns that were raised, but they provided additional information and clarifications that were helpful in understanding the behavior of mice in their interesting and complex learning paradigm. The authors also acknowledged that there were questions raised in my review that could not be addressed by their data. This latter point does not concern me because it is not usual for this to be the case in the early development of new paradigms. Finally, the authors were able to demonstrate quite rapid relearning in transfer that resembles rapid rule transfer which is a very interesting addition to the paper. Based on the reviewers' revision, in my opinion the manuscript has been sufficiently improved so that it now meets criterion for publication in Nature Communications, so I recommend that it be accepted for publication.

Reviewer #2 (Remarks to the Author):

The authors have added new analysis and new experiments that fully addressed my concerns. I believe that the revised manuscript is much improved and suitable for publication.

I have a few comments regarding the revised version.

The authors added to the discussion new results that are quite interesting but that I feel should be part of the result section.

- 1) The section in the discussion that reports the position of the opponent while one mouse is getting reward should be placed after the results section on the evolution of territory (maybe after line 169).
- 2) The section on the persistence of rule observance behavior (lines 296-303) could correspond to the last paragraph of the result section.

The paragraph in the discussion on the limitations of this study (304-313) is a list of points made by the reviewers that this study does not address. I believe this paragraph does not really add to the paper. The authors did a better job at discussing these issues in the rebuttal letter. I believe that, the authors should discuss the relevance of the issues raised and why their study does not address them (even if briefly) or remove this section of the discussion.

Nature Communications revision for MS# NCOMMS-17-01018-A

Title: Mice in conflict show rule-observance behavior enhancing long-term benefit

Point-by-point responses to referees' comments:

We have tried to revise the manuscript based on the valuable comments. The positions of the revised text in response to the comments are indicated by 'Line #~#' in this response sheet and are highlighted in the text.

Reviewer #1 (Remarks to the Author):

The authors have made a great effort to address all concerns raised by the reviewers. With regard to criticisms raised by this reviewer, the authors were not able to address all concerns that were raised, but they provided additional information and clarifications that were helpful in understanding the behavior of mice in their interesting and complex learning paradigm. The authors also acknowledged that there were questions raised in my review that could not be addressed by their data. This latter point does not concern me because it is not usual for this to be the case in the early development of new paradigms. Finally, the authors were able to demonstrate quite rapid relearning in transfer that resembles rapid rule transfer which is a very interesting addition to the paper. Based on the reviewers' revision, in my opinion the manuscript has been sufficiently improved so that it now meets criterion for publication in Nature Communications, so I recommend that it be accepted for publication.

We deeply appreciate the reviewers for critical and constructive comments, which have helped us improve the quality of our manuscript tremendously.

Reviewer #2 (Remarks to the Author):

The authors have added new analysis and new experiments that fully addressed my concerns. I believe that the revised manuscript is much improved and suitable for publication.

I have a few comments regarding the revised version.

The authors added to the discussion new results that are quite interesting but that I feel should be part of the result section.

1) The section in the discussion that reports the position of the opponent while one mouse is getting reward should be placed after the results section on the evolution of territory (maybe

after line 169).

We agree with the reviewer's comment and placed the paragraph in the result section. (lines 227 - 240)

2) The section on the persistence of rule observance behavior (lines 296-303) could correspond to the last paragraph of the result section.

We covered the section on the persistent rule-observance in the last paragraph of the result section. (lines 267 – 272)

The paragraph in the discussion on the limitations of this study (304-313) is a list of points made by the reviewers that this study does not address. I believe this paragraph does not really add to the paper. The authors did a better job at discussing these issues in the rebuttal letter. I believe that, the authors should discuss the relevance of the issues raised and why their study does not addressed them (even if briefly) or remove this section of the discussion.

We removed the paragraph in the discussion section about limitations of our study as recommended. (lines 315 – 322)

We are thankful for the reviewer's recommendation that improved our manuscript tremendously.